# Convergence of Adafactor under Non-Convex Smooth Stochastic Optimization

## Abstract

As model sizes in deep learning continue to expand, memory-efficient optimizers are increasingly critical to manage the substantial memory demands of popular algorithms like Adam and AdamW. Among these, Adafactor has emerged as one of the widely adopted choices for training deep learning tasks, particularly large language models. However, despite its practical success, there is limited theoretical analysis on Adafactor's convergence. This paper presents a comprehensive analysis on Adafactor in a non-convex smooth setting, demonstrating its convergence to find a stationary point at a rate of $\tilde{\mathcal{O}}(1/\sqrt{T})$. We find that the default hyper-parameter setting results in a sub-optimal rate in our framework, and propose an alternative setting that could theoretically achieve optimal convergence rate. This finding is further supported by some experimental results. We also prove that Adafactor with a suitable time-varying clipping threshold could also converge, achieving performance in experiments comparable to that of the standard constant setting.

## 1 Introduction

The adaptive gradient-based methods, such as the well-known AdaGrad [9, 29], RMSProp [30], Adadelta [35], Adam [15] and AdamW [22], have become the preferred approaches in solving the following unconstrained stochastic optimization problem in deep learning fields:

$$\min_{\boldsymbol{X} \in \mathbb{R}^{n \times m}} f(\boldsymbol{X}) = \mathbb{E}_{\boldsymbol{Z} \in \mathcal{P}}[l(\boldsymbol{X}; \boldsymbol{Z})], \tag{1}$$

where the object function $f$ is non-convex and $\mathcal{P}$ denotes a probability distribution. During the training process, these adaptive methods require to store the historical gradients' information so as to adaptively tune their step-sizes. For example, both Adam and AdamW maintain the exponential average of gradients and squared gradients, and AdaGrad stores the cumulative of squared gradients. Despite their effectiveness, these algorithms pose substantial memory challenges for GPUs to save these additional gradients' information, especially when training large language models (LLMs), such as GPT-3 [5], which contains over 175 billion parameters.

To address memory constraints, several memory-efficient optimization algorithms have been developed, e.g., [26, 1, 23, 17]. One of the most popular optimizers is Adafactor [26] which employs a rank-1 matrix factorization to approximate the second moment matrix in Adam. For an $n \times m$ weight matrices, this technique reduces memory usage from $\mathcal{O}(mn)$ to $\mathcal{O}(m + n)$ by only tracking the moving averages of the row and column sums of the squared gradients matrix. Additionally, Adafactor eliminates the first-order momentum used in Adam and incorporates update clipping to enhance training stability.

The empirical results reveal that Adafactor achieves comparable performance to Adam in training Transformer models [26] . In real applications, several LLMs including PaLM [6] and T5 [24] have

applied Adafactor as their main optimizers [38]. In spite of Adafactor's widely usage, there is still limited understanding on its convergence in theory, especially the effect of the matrix approximation and update clipping, and the explanation for its hyper-parameter setting in experiments.

In this paper, we take a closer look on Adafactor's convergence under non-convex smooth optimization problems, considering the typical bounded gradient setting as those for AdaGrad [19, 32] and Adam [34]. We aim to provide a convergence rate for Adafactor and explain the influence of the hyper-parameters for the convergence speed. We also prove in theory why the default parameter setting is effective in practical scenarios. The analysis to Adafactor is non-trivial compared to other adaptive methods such as AdaGrad and Adam due to the unique matrix factorization and update clipping mechanisms. Based on a new proxy step-size construction and some new compositions as well as estimations, we analyze the additional error terms in the Descent Lemma introduced by the matrix approximation and update clipping. Our main contributions are summarized as follows.

**Contributions**

- We provide a convergence analysis for the full-batch Adafactor considering bounded gradients and a broader range of parameter setting which covers the default one in [26]. The result shows that Adafactor could converge to find a stationary point with a rate of $\tilde{\mathcal{O}}(1/\sqrt{T})$ where $T$ denotes the total iteration number.

- We further investigate the more realistic stochastic Adafactor. It's found that a simple variant of Adafactor, which drops the update clipping, could attain the best convergence rate of $\tilde{\mathcal{O}}(1/\sqrt{T})$ when the second moment decay rate is $1 - 1/k$. We also verify that the default decay rate $1 - 1/k^{0.8}$ could lead to a sub-optimal convergence rate in our framework. To illustrate this finding, we provide some empirical results, showing that the potential best hyper-parameter setting in theory could perform better than the default one used in experiments.

- We extend our study to include a time-varying clipping threshold. Our analysis implies that with proper selections of clipping threshold and hyper-parameters, Adafactor could also achieve the best convergence rate of $\tilde{\mathcal{O}}(1/\sqrt{T})$. We also do some experiments to show that the new clipping threshold scheme achieves comparable performance and training stability to the original constant threshold setting.

The rest of the paper is organized as follows. The next section provides some most relevant works. Section 3 presents some necessary notations definitions and problem setup. Section 4 reviews Adafactor and introduces its essential mechanism. In Section 5 and Section 6, we separately provide convergence bounds for full-batch Adafactor and stochastic Adafactor without update clipping. We further discuss the hyper-parameters' dependency. In Section 7, we investigate Adafactor using a time-increasing update clipping threshold. Section 8 provides experimental results to support our theory. All the detailed proof could be found in the appendix.

## 2  Related work

In this paper, we mainly investigate the theoretical convergence of Adafactor. Although there is limited works on Adafactor in theory, it's necessary to briefly discuss related works on the convergence of other adaptive methods, particularly on non-convex smooth optimization. Here, we briefly list some of the most related works.

**Convergence of adaptive methods**  Several studies address the convergence of AdaGrad in non-convex settings. For example, [19] considered a simple variant with delayed step-size, while [32] and [39] assumed bounded stochastic gradients. Other works [14, 10, 21, 3, 31, 27, 33] derived convergence bounds under more relaxed assumptions. Another line of research has investigated the convergence of Adam. For instance, [34, 7, 39, 11, 8] assumed bounded gradients. [28, 36, 31] considered more relaxed noise assumptions without relying on bounded gradients. Additionally, [18] derived convergence bounds for Adam under generalized smooth conditions.

Overall, the convergence analysis of optimizers typically starts with standard assumptions, such as bounded gradients and smooth objective functions. In subsequent studies, these assumptions are gradually relaxed to investigate the convergence properties of the optimizers under less stringent conditions.

**Memory efficient algorithms**   As large models are increasingly used in deep learning, memory constraints have become a central issue during training. Consequently, several memory-efficient optimizers have been developed to address this challenge.

One approach to save memory involves applying matrix factorization to oeptimization algorithms. For instance, [25] used matrix factorization in the second moment estimator of gradients in Adam, similar to the concept behind Adafactor. [23] introduced CAME, a variant of Adafactor, which incorporates a confidence-guided strategy to mitigate instability caused by erroneous updates. [37] proposed Adapprox, leveraging randomized low-rank matrix approximation for Adam's second moment estimator, demonstrating superior performance and reduced memory usage compared to AdamW.

There are some other techniques to save the memory. For example, [12] relied on a "Shampoo" technique to reduce the storage requirement of full-matrix preconditioning methods. Notably, their method could be further extended to the more realistic tensor case. [1] presented a memory-saved version of AdaGrad, called SM3, by maintaining $k$ sets gradient accumulator. They proved the convergence guarantee of SM3 on online convex optimization and the effectiveness in experiments. Recently, [17] built a 4-bit Adam using quantization techniques to compress the first and second moment estimators in Adam, also reducing memory usage.

In summary, many existing optimizers, particularly adaptive methods like AdaGrad and Adam, face memory overhead. In response, the discussed works have designed memory-efficient optimizers that aim to achieve comparable performance to these existing methods while achieving memory benefits.

## 3   Problem setup

To start with, we introduce some necessary notations.

**Notations**   For any two matrices $\boldsymbol{X} = (x_{ij})_{ij}, \boldsymbol{Y} = (y_{ij})_{ij} \in \mathbb{R}^{n \times m}$, we define $\langle \boldsymbol{X}, \boldsymbol{Y} \rangle = \sum_{i=1}^{n} \sum_{j=1}^{m} x_{ij} y_{ij}$. $\boldsymbol{X} \odot \boldsymbol{Y}$, $\boldsymbol{X}/\boldsymbol{Y}$ and $\sqrt{\boldsymbol{X}}$ denote the coordinate-wise product, quotient and squared root respectively. $\boldsymbol{0}_n$ and $\boldsymbol{1}_n$ denote the zero and one $n$-dimensional vector respectively, and $\boldsymbol{1}_{n \times m}$ denotes the one $n \times m$-dimensional matrix. The index set $[n]$ denotes $\{1, 2, \cdots, n\}$. $\| \cdot \|_F$ denotes the Frobenius norm. For a positive sequence $\{\alpha_i\}_{i \geq 1}$, we define $\sum_{i=a}^{b} \alpha_i = 0$ and $\prod_{i=a}^{b} \alpha_i = 1$ if $a > b$. The operator $\mathrm{RMS}(\cdot)$ denotes

$$\mathrm{RMS}(\boldsymbol{X}) = \sqrt{\frac{1}{mn} \sum_{i=1}^{n} \sum_{j=1}^{m} x_{ij}^2}.$$

We consider unconstrained stochastic optimization (1) over $\mathbb{R}^{n \times m}$ with the Frobenius norm. The objective function $f : \mathbb{R}^{n \times m} \to \mathbb{R}$ is differentiable. Given an $n \times m$ matrix $\boldsymbol{X}$, we assume a gradient oracle that returns a random matrix $g(\boldsymbol{X}, \boldsymbol{Z}) \in \mathbb{R}^{n \times m}$ dependent by the random sample $\boldsymbol{Z}$. The deterministic gradient of $f$ at $\boldsymbol{X}$ is denoted by $\nabla f(\boldsymbol{X}) \in \mathbb{R}^{n \times m}$.

**Assumptions**   We make the following standard assumptions throughout the paper.

- **(A1)** $L$-smoothness: For any $\boldsymbol{X}, \boldsymbol{Y} \in \mathbb{R}^{n \times m}$, $\|\nabla f(\boldsymbol{Y}) - \nabla f(\boldsymbol{X})\|_F \leq L\|\boldsymbol{Y} - \boldsymbol{X}\|_F$;
- **(A2)** Bounded below: There exists $f^* > -\infty$ such that $f(\boldsymbol{X}) \geq f^*, \forall \boldsymbol{X} \in \mathbb{R}^{n \times m}$;
- **(A3)** Unbiased estimator: The gradient oracle provides an unbiased estimator of $\nabla f(\boldsymbol{X})$, i.e., $\mathbb{E}_{\boldsymbol{Z}}[g(\boldsymbol{X}, \boldsymbol{Z})] = \nabla f(\boldsymbol{X}), \forall \boldsymbol{X} \in \mathbb{R}^{n \times m}$;
- **(A4)** Almost surely bounded stochastic gradient: for any $\boldsymbol{X} \in \mathbb{R}^{n \times m}$, $\|g(\boldsymbol{X}, \boldsymbol{Z})\|_F \leq G$, a.s..

Combining (A3) and (A4), it's easy to verify that $\|\nabla f(\boldsymbol{X})\| \leq G, \forall \boldsymbol{X} \in \mathbb{R}^{n \times m}$. Assumptions (A1)-(A3) are standard in the non-convex smooth convergence analysis. Although Assumption (A4) is a bit strong since it requires an almost surely bounded stochastic gradients instead of an expected one, it's still commonly used to derive the high probability convergence bound, see e.g., [32, 14], which is a stronger result than an expected convergence. In coordinate-wise algorithm, another standard assumption is $l_\infty$-bounded gradient where $\|g(\boldsymbol{X}, \boldsymbol{Z})\|_\infty \leq G_\infty$, see e.g., [8]. These two types of assumption are equivalent up to dimension factors.

## 4 Review of Adafactor

In this section, we briefly discuss Adafactor based on the reference [26]. The pseudocode for Adafactor is presented in Algorithm 1.

---

**Algorithm 1** Adafactor

---

**Input:** Initialization point $\boldsymbol{X}_1 \in \mathbb{R}^{n \times m}$, $\boldsymbol{R}_0 = \boldsymbol{0}_m$, $\boldsymbol{C}_0 = \boldsymbol{0}_n^\top$, relative step-sizes $\{\rho_k\}_{k \geq 1}$, decay rate $\{\beta_{2,k}\}_{k \geq 1} \in [0, 1)$, regularization constants $\epsilon_1, \epsilon_2 > 0$, clipping threshold $d$.
  **for** $k = 1, \cdots, T$ **do**
    $\boldsymbol{G}_k = g(\boldsymbol{X}_k, \boldsymbol{Z}_k)$;
    $\boldsymbol{R}_k = \beta_{2,k} \boldsymbol{R}_{k-1} + (1 - \beta_{2,k})(\boldsymbol{G}_k \odot \boldsymbol{G}_k + \epsilon_1 \boldsymbol{1}_n \boldsymbol{1}_m^\top) \boldsymbol{1}_m$;
    $\boldsymbol{C}_k = \beta_{2,k} \boldsymbol{C}_{k-1} + (1 - \beta_{2,k}) \boldsymbol{1}_n^\top (\boldsymbol{G}_k \odot \boldsymbol{G}_k + \epsilon_1 \boldsymbol{1}_n \boldsymbol{1}_m^\top)$;
    $\boldsymbol{W}_k = (\boldsymbol{R}_k \boldsymbol{C}_k) / \boldsymbol{1}_n^\top \boldsymbol{R}_k$;
    $\boldsymbol{U}_k = \boldsymbol{G}_k / \sqrt{\boldsymbol{W}_k}$;
    $\eta_k = \max\{\epsilon_2, \text{RMS}(\boldsymbol{X}_k)\} \rho_k / \max\{1, \text{RMS}(\boldsymbol{U}_k)/d\}$;
    $\boldsymbol{X}_{k+1} = \boldsymbol{X}_k - \eta_k \cdot \boldsymbol{G}_k / \sqrt{\boldsymbol{W}_k}$;
  **end for**

---

**Matrix factorization**   Adafactor could be severed as a saved-memory version of Adam. Throughout the training process, Adam maintain two $n \times m$ matrices $\boldsymbol{M}_k$ and $\boldsymbol{V}_k$ using exponential moving average update,

$$\boldsymbol{M}_k = \beta_{1,k} \boldsymbol{M}_{k-1} + (1 - \beta_{1,k}) \boldsymbol{G}_k, \quad \boldsymbol{V}_k = \beta_{2,k} \boldsymbol{V}_{k-1} + (1 - \beta_{2,k}) \boldsymbol{G}_k \odot \boldsymbol{G}_k, \tag{2}$$

where $\beta_{1,k}, \beta_{2,k} \in (0, 1)$, thereby tripling the memory usage. The innovation in Adafactor lies in its method of approximating $\boldsymbol{V}_k$ by factoring it into two rank-1 matrices, specifically the row sums and column sums of $\boldsymbol{V}_k$. This approximation is guided by maintaining a minimal general Kullback-Leibler (KL) divergence as follows,

$$\min_{\boldsymbol{X} \in \mathbb{R}^n, \boldsymbol{Y} \in \mathbb{R}^{1 \times m}} \sum_{i=1}^{n} \sum_{j=1}^{m} d\left( (\boldsymbol{V}_k)_{ij}, (\boldsymbol{XY})_{ij} \right), \quad \text{s.t.} \quad (\boldsymbol{X})_i \geq 0, (\boldsymbol{Y})_j \geq 0, \forall i \in [n], j \in [m],$$

where $d(p, q) = p \log(p/q) - p + q$. The choice of KL-divergence over the more typical Frobenius norm allows for an analytical solution to be derived, specifically given by

$$\boldsymbol{X} = \boldsymbol{V}_k \boldsymbol{1}_m, \quad \boldsymbol{Y} = \boldsymbol{1}_n^\top \boldsymbol{V}_k / \left( \boldsymbol{1}_n^\top \boldsymbol{V}_k \boldsymbol{1}_m \right).$$

Therefore, Adafactor only requires to maintain two vectors $\boldsymbol{R}_k = \boldsymbol{V}_k \boldsymbol{1}_m$, $\boldsymbol{C}_k = \boldsymbol{1}_n^\top \boldsymbol{V}_k$, sufficiently reducing the memory from $2mn$ to $m + n$. Although this factorization sacrifices some information of the squared gradients, Adafactor still delivers performance comparable to Adam in many real application tasks, making it a practical choice where memory is a constraint.

**Increasing decay rate**   In Adam, corrective terms are introduced into $\boldsymbol{M}_k$ and $\boldsymbol{V}_k$, resulting in two increasing-to-one decay rates. Theoretically, it has been demonstrated that a value closed to one for $\beta_{2,k}$ would ensure the convergence, e.g., [8, 39, 36]. Inspired by this observation, Adafactor used an increasing second moment decay rate $\beta_{2,k} = 1 - 1/k^c, c \geq 0$, and the empirical default setting is $c = 0.8$. As pointed out by [26], this setting allows for enjoying the stability of a low $\beta_{2,k}$ at the early stage of training and the insurance of convergence from a high $\beta_{2,k}$ as the run progresses. Moreover, it also leverages the bias correction.

**Update clipping**   Adafactor modifies the update process by discarding the first-order moment $\boldsymbol{M}_k$ and instead applies an update clipping technique inside the step-size $\eta_k$. This involves dividing the root-mean-square of the update $\boldsymbol{U}_k$, denoted as $\text{RMS}(\boldsymbol{U}_k)$, when it exceeds a threshold $d$. This mechanism helps to calibrate the second moment estimator $\boldsymbol{W}_k$ when it's larger-than-desired $\boldsymbol{G}_k \odot \boldsymbol{G}_k$. Empirical findings in [26] indicated that implementing update clipping leads to significant performance improvements when the warm-up technique is not used.

**Relative step-sizes**   Adafactor incorporates a step-size proportional to scale of $\boldsymbol{X}_k$, denoted by $\text{RMS}(\boldsymbol{X}_k)$, which is shown in experiments more resilient to the more naive parameter initialization and scaling schemes [26].

## 5  Convergence result for full-batch Adafactor

We first provide the convergence bound for full-batch Adafactor. At each iteration, full-batch Adafactor obtains the deterministic gradient $\nabla f(\boldsymbol{X}_k)$ and then updates $\boldsymbol{R}_k, \boldsymbol{C}_k$ using $\nabla f(\boldsymbol{X}_k)$ instead of $\boldsymbol{G}_k$ in Algorithm 1.

**Theorem 5.1.** *Let $\{\boldsymbol{X}_k\}_{k\geq 1}$ be generated by Algorithm 1 with $g(\boldsymbol{X}_k, \boldsymbol{Z}_k) = \nabla f(\boldsymbol{X}_k), \forall k \geq 1$. If Assumptions (A1) and (A2) hold, $\|\nabla f(\boldsymbol{X}_k)\|_F \leq G, \forall k \geq 1$, $\beta_{2,1} = \frac{1}{2}$ and*

$$\rho_k = \rho_0/\sqrt{k}, \quad 0 < \beta_{2,k} < 1, \quad \forall k \geq 1, \tag{3}$$

*for some positive constant $\rho_0$, then for any $T \geq 1$,*

$$\min_{k\in[T]} \|\nabla f(\boldsymbol{X}_k)\|_F^2 \leq \mathcal{O}\left(\frac{\log T}{\sqrt{T}}\right). \tag{4}$$

The result indicates that full-batch Adafactor could find a stationary point at a rate of $\mathcal{O}(\log T/\sqrt{T})$ under the non-convex smooth case, which is similar to gradient descent but with a sub-optimal rate compared to $\mathcal{O}(1/T)$ [4]. The hyper-parameter setting in (3) only requires $\beta_{2,k} \in (0, 1)$, denoting a much wider range including the default one which requires $\beta_{2,k}$ to increase to one. The detailed version for the above result can be found in Theorem A.1 from the appendix.

## 6  Stochastic Adafactor without update clipping

In the stochastic case, we start from the simple scenario of

$$\eta_k = \max\{\epsilon_2, \text{RMS}(\boldsymbol{X}_k)\}\rho_k \tag{5}$$

without considering the update clipping $1/\max\{1, \text{RMS}(\boldsymbol{U}_k)/d\}$ in Algorithm 1, where the main reasons are as follows.

- As pointed out in the experiments from [26], Adafactor's performance shows little difference with and without update clipping when implementing learning rate warm-up. Since the warm-up technique is a popular method in deep learning [38], it's reasonable to drop the update clipping.
- In stochastic Adafactor, the correlation between $\boldsymbol{G}_k$ and $\eta_k$ would be more complex if the update clipping is involved. The proof would be simpler when dropping the update clipping, which could help to better understand the analysis for Adafactor.

We now present the probabilistic convergence bound for Adafactor without update clipping as follows, where we summarize different convergence rate with respect to the factor $c$ from $\beta_{2,k} = 1 - 1/k^c, c \in [1/2, 1]$.

**Theorem 6.1.** *Let $\{\boldsymbol{X}_k\}_{k\geq 1}$ be generated by Algorithm 1 without update clipping where $\eta_k$ is given by (5) for each $k \geq 1$. If Assumptions (A1)-(A4) hold, and*

$$\begin{aligned}
\beta_{2,1} &= 1/2, \quad \rho_1 = \rho_0, \\
\beta_{2,k} &= 1 - 1/k^c, \quad \rho_k = \rho_0/\sqrt{k}, \quad \forall k \geq 2,
\end{aligned} \tag{6}$$

*for some constants $1/2 \leq c \leq 1, \rho_0 > 0$, then for any $T \geq 1, \delta \in (0, 1)$, with probability at least $1 - \delta$,*

$$\min_{k\in[T]} \|\nabla f(\boldsymbol{X}_k)\|_F^2 \leq \mathcal{O}\left(\frac{1}{T^{c-1/2}} \log\left(\frac{T}{\delta}\right)\right).$$

The above result indicates that with appropriate hyper-parameters, Adafactor without update clipping could approximately find a stationary point. When the decay rate $\beta_{2,k}$ is $1 - 1/k$, the convergence rate could attain to $\mathcal{O}(\log T/\sqrt{T})$, matching the rate of stochastic gradient descent [4] and the lower rate in [2] up to only a logarithm factor. The hyper-parameter setting in (6) covers the experimental default setting where $c = 0.8$. The result shows a sub-optimal rate of $\mathcal{O}(\log T/T^{0.3})$ under the default setting. This finding is further complemented by the coming numerical experiments in Section 8. The detailed version of the above results can be found in Theorem B.1 from the appendix.

## 6.1 Discussion of the hyper-parameter dependency

In this section, we discuss the dependency of several important hyper-parameters in Theorem 6.1 and the detailed version in Theorem B.1 in the appendix. It's worthy to mention that the dominated order in our convergence bound is determined by the total iteration number $T$, whereas other hyper-parameters could be regarded as constants. However, we hope to improve the dependency of these hyper-parameters as much as possible to make the convergence bound tight.

**Discussion of $c$ and the optimal rate** The convergence bound in Theorem 6.1 reveals that when $c = 1, \beta_{2,k} = 1 - 1/k$ and $\rho_k = \rho_0/\sqrt{k}$, the convergence rate attains the optimal rate matching the lower bound. In addition, when $c$ is closed to $1/2$, the convergence rate deteriorates. This phenomenon somehow explains that a small decay rate $\beta_{2,k}$ ($c$ is low) may harm the convergence speed, as $\beta_{2,k}$ should be closed enough to 1 to ensure the convergence, which has been pointed out similarly for Adam in [8, 39, 36].

The theoretical best parameter setting remains a small gap to the default one of $c = 0.8$. To verify our theoretical finding, we provide some empirical evidence in Section 8, showing that $\beta_{2,k} = 1 - 1/k$ performs even better than the default one and the performance would be better when $c$ increases from $1/2$ to 1.

**Dependency to $mn$** It's clear to see that the convergence bounds in Theorem A.1 and Theorem B.1 are free of the curse of the dimension factor $mn$ as $mn$ only appears on the denominator in each coefficient. We think that solving the curse of dimension is vital since the applied range for Adafactor includes many deep learning tasks where $mn$ are comparable large to $T$.

**Dependency to $\epsilon_1, \epsilon_2$** The convergence bounds in (37) and (39) from Theorem B.1 has a dependency of $\mathcal{O}(\epsilon_1^{-1}\log(1/\epsilon_1))$ on $\epsilon_1$.[1] Although the polynomial dependency to $\epsilon_1$ is a bit worse since $\epsilon_1$ ususally takes a small value in experiments, e.g., the default setting is $10^{-30}$, it's still common in some theoretical convergence results, e.g., [34, 18]. We also perform some experiments to show that a relatively large $\epsilon_1$, roughly $10^{-3}$, makes no observable effect on the performance. Thereby, $\epsilon_1$ could be regarded as a constant in comparison to $T$ and the influence brought by $1/\epsilon_1$ could be somehow acceptable.

Since the default value of $\epsilon_2$ is $10^{-3}$ in experiments, it could also be regarded as a constant compared to $T$. Therefore, the dependency $\mathcal{O}(1/\epsilon_2)$ on $\epsilon_2$ shows little effect on convergence bounds given the sufficiently large $T$.

**Dependency to the scale of parameters.** The convergence bounds in Theorem B.1 contain a $\mathcal{O}(\Theta_{\max})$ factor where $\Theta_{\max}$ denotes the maximum values of $\|\boldsymbol{X}_k\|_\infty$ along the training process. It's reasonable to assume that $\Theta_{\max} \leq G_0$ for a comparable large constant $G_0$ in practice.

## 7 Convergence of Adafactor with update clipping

In this section, we take a closer look on the comprehensive Adafactor with both matrix factorization and update clipping. We slightly change the update clipping threshold $d$ in Algorithm 1 to a time-varying threshold $d_k$. The step-size $\eta_k$ then becomes

$$\eta_k = \frac{\max\{\epsilon_2, \mathrm{RMS}(\boldsymbol{X}_k)\}\rho_k}{\max\{1, \mathrm{RMS}(\boldsymbol{U}_k)/d_k\}}. \tag{7}$$

Then, we present the following convergence bound.

**Theorem 7.1.** *Let $\{\boldsymbol{X}_k\}_{k\geq 1}$ be generated by Algorithm 1 with $\eta_k$ given by (7) for each $k \geq 1$. If Assumptions (A1)-(A4) hold, and*

$$
\begin{aligned}
d_1 &= 1, \quad \beta_{2,1} = 1/2, \quad \rho_1 = \rho_0, \\
d_k &= k^{\frac{c}{2(\alpha-1)}}, \quad \beta_{2,k} = 1 - 1/k^c, \quad \rho_k = \rho_0/\sqrt{k}, \quad \forall k \geq 2,
\end{aligned}
\tag{8}
$$

---

[1] The detailed calculation could be found in (45) and (46) in the appendix.

*for some constants $\alpha > 1, 1/2 \leq c \leq 1, \rho_0 > 0$, then for any $T \geq 1, \delta \in (0, 1)$, with probability at least $1 - \delta$,*

$$\min_{k \in [T]} \|\nabla f(\boldsymbol{X}_k)\|_F^2 \leq \mathcal{O}\left(\frac{1}{T^{c-1/2}} \log\left(\frac{T}{\delta}\right)\right).$$

**Discussion of Theorem 7.1**    The convergence result indicates that with a proper selection of the clipping threshold, along with the commonly used step-size $\rho_k$ and decay rate $\beta_{2,k}$, Adafactor can find a stationary point when $T$ is sufficiently large. The dependency of convergence bound on $c$ remains consistent with Theorem 6.1, achieving the optimal order when $c = 1$. In addition, the convergence bound can still avoid the curse of dimension, which is shown in the detailed version Theorem C.1 from the appendix.

The additional hyper-parameter $\alpha$ primarily influences the dependency on $\epsilon_1$, specifically as $\mathcal{O}\left(\epsilon_1^{-\alpha} \log(1/\epsilon_1)\right)$. Thus, our convergence bound may deteriorate as $\alpha$ increases, possibly due to the limitation of our proof framework. This dependency could be potentially improved to $\mathcal{O}\left(\epsilon_1^{-1} \log(1/\epsilon_1)\right)$ when $mn$ is comparable to $1/\epsilon_1$, which is practical when implementing a large-size model.[2] In our experiments, we tested different values of $\alpha$ and found that suitably small values, such as $\alpha = 4, 6, 7, 8$ can lead to performance and training stability comparable to the default setting, even without implementing the warm-up technique. This finding suggests that our new threshold setting plays a similar role in enhancing training stability as the default one, which is also the main motivation of update clipping. Since $\epsilon_1$ can be set to a relatively large value, e.g., $10^{-3}$, a dependency like $\mathcal{O}(\epsilon_1^{-4} \log(1/\epsilon_1))$ is somewhat acceptable for sufficiently large $T$.

The time-increasing $d_k$ provides the following intuition: As shown in [26, Figure 1], during the early stages of training, a high decay rate $\beta_{2,k}$ can cause larger-than-desired updates and training instability. Therefore, we set a low threshold $d_k$ to ensure that the update clipping mechanism effectively calibrates these larger-than-desired updates. As training progresses, the sequences and updates become more stable, and the second moment estimator $\boldsymbol{W}_k$ becomes more accurate in estimating the squared gradients, which is also shown in [26, Figure 1]. Consequently, there is less need for update clipping, corresponding to a relatively large $d_k$. We have also verified through experiments that our setting can achieve performance comparable to the default setting of $d = 1$.

# 8  Experiments

In this section, we will report our experimental results based on the insights obtained in our theory. We will mainly provide the following three experiments:

- We test Adafactor without update clipping under different decay rate parameters $c$, aiming to demonstrate performance improvement as $c$ increases from $0.5$ to $1$ with optimal performance at $c = 1$, as indicated in Theorem 6.1 and Theorem 7.1.

- We evaluate the sensitivity of Adafactor to different values of $\epsilon_1$, particularly showing that a relatively large $\epsilon_1$ does not significantly impact performance.

- We assess the performance of Adafactor with a time-increasing $d_k$ setting, as described in Theorem 7.1, and compare it to the default constant setting.

## 8.1  Experiment setup

In all experiments, the initialization is $\boldsymbol{R}_0 = \boldsymbol{0}_m$ and $\boldsymbol{C}_0 = \boldsymbol{0}_n^\top$. We use a learning rate with the warm-up technique as described in [26], specifically $\rho_k = \min\{10^{-6} \cdot k, 1/\sqrt{k}\}$ for all experiments unless otherwise specified. The batch size is set to 256, and the total number of epochs is 400 by default. Our models are ResNet-20 and ResNet-110 [13], and we use the CIFAR-10 and CIFAR-100 datasets [16] without any data augmentation. The experiments are conducted using the PyTorch implementation of Adafactor on a single NVIDIA GeForce RTX 4090 GPU.

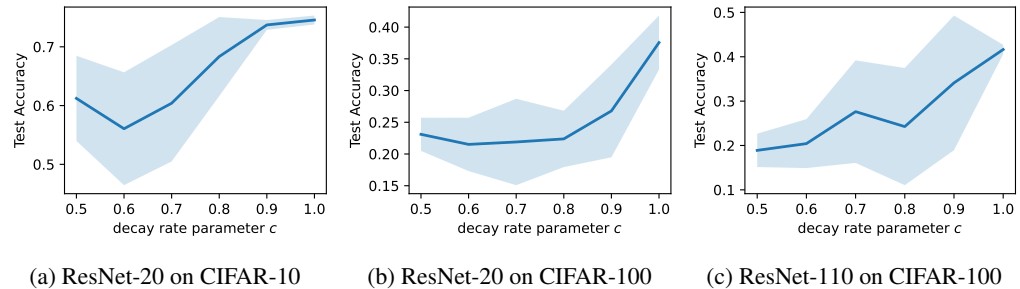

(a) ResNet-20 on CIFAR-10  (b) ResNet-20 on CIFAR-100  (c) ResNet-110 on CIFAR-100

Figure 1: Average test accuracy and standard deviation (shallow blue region) under different decay rate parameters $c$.

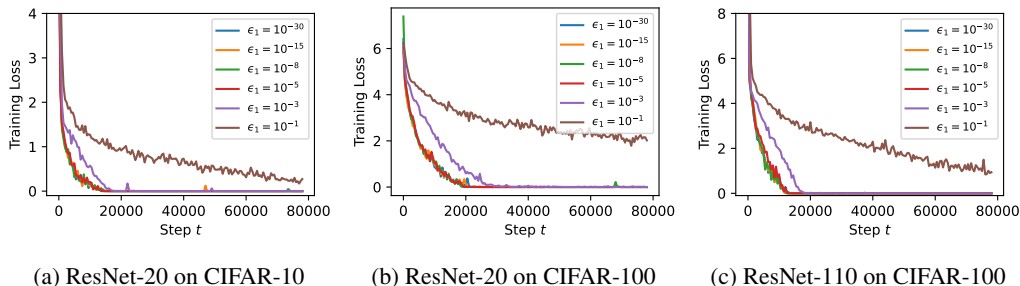

(a) ResNet-20 on CIFAR-10  (b) ResNet-20 on CIFAR-100  (c) ResNet-110 on CIFAR-100

Figure 2: Training loss vs. steps using Adafactor without update clipping under different $\epsilon_1$. The step-size $\eta_t$, decay rate $\beta_{2,k}$, and learning rate warm-up are set by default.

## 8.2 Report on Experiment 1

We test Adafactor without update clipping using decay rate parameter $c$ ranging from $0.5$ to $1.0$ in increments of $0.05$, while keeping other hyper-parameters at their default values. Each experiment is run 10 times with 100 epochs, and we plot the average test accuracy and standard deviation (shallow blue region) in Figure 1. The results indicate that $c = 1.0$ yields better performance and stability compared to $c < 1.0$ on different models and datasets, corresponding to the highest test accuracy and thinner shallow blue band. These performances show a noticeable improving trend as $c$ increases from $0.5$ to $1.0$, aligning roughly with the results in Theorem 6.1.

## 8.3 Report on Experiment 2

In the second experiment, we test Adafactor without update clipping under different $\epsilon_1$ values. We plot the training loss against the step $t$ on different models and datasets in Figure 2. The performance for $\epsilon_1 = 10^{-8}$ and $\epsilon_1 = 10^{-5}$ is nearly identical to that for $\epsilon_1 = 10^{-30}$. Moreover, even a larger value of $10^{-3}$ achieves comparable training performance, though with a slower decrease in loss. Notably, $\epsilon_1 = 10^{-3}$ requires approximately the same number of steps ($t \approx 20000$) as $\epsilon_1 = 10^{-30}$ to achieve near-zero training loss. We conclude that Adafactor is not sensitive to the choice of $\epsilon_1$, and a relatively large $\epsilon_1$ can still lead to convergence, making the polynomial dependency $\mathcal{O}(1/\epsilon_1)$ in our convergence bounds acceptable.

## 8.4 Report on Experiment 3

In this experiment, we explore the appropriate values of $\alpha$ in Theorem 7.1 to achieve performance comparable to the default setting of $d = 1$. As indicated by Theorem 7.1, a relatively small $\alpha$ is desirable for better dependency on $\epsilon_1$. We train models with $\alpha$ set to 4, 6, 7, 8, and 9, keeping other hyper-parameters at their default values. We also train models with the default $d = 1$ setting as the baseline. We plot the training loss against the steps in Figures 3 without step-size warm-up and 4 with step-size warm-up.

---

[2] The detailed calculation could be found in (95) from the appendix.

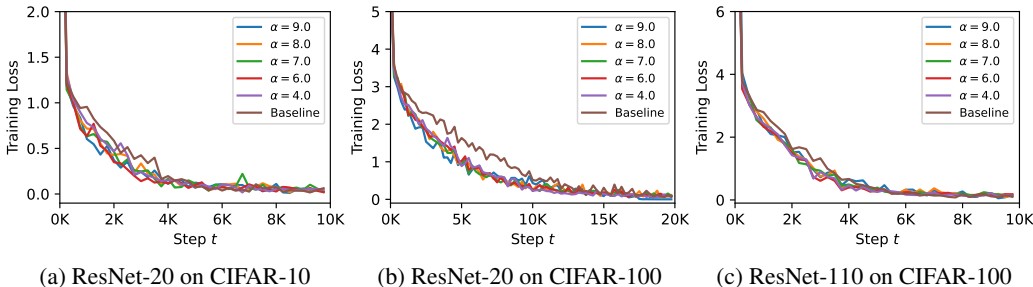

(a) ResNet-20 on CIFAR-10     (b) ResNet-20 on CIFAR-100     (c) ResNet-110 on CIFAR-100

Figure 3: Training loss vs. steps on different models and datasets. We use step-size without warm-up technique and test under different $\alpha$.

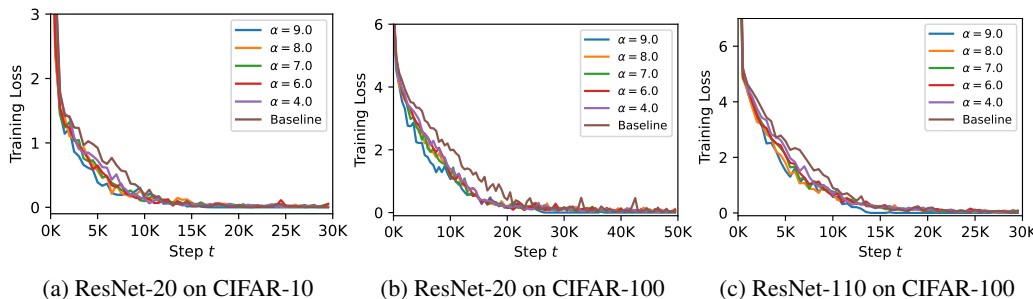

(a) ResNet-20 on CIFAR-10     (b) ResNet-20 on CIFAR-100     (c) ResNet-110 on CIFAR-100

Figure 4: Training loss vs. steps on different models and datasets. We use step-size with warm-up technique by default and test under different $\alpha$.

The results indicate that, for these values of $\alpha$, Adafactor achieves comparable or even better convergence speed compared to the default threshold (represented by "Baseline"). The comparable results to the "Baseline" in Figure 3 further suggest that the time-increasing $d_k$ in Theorem 7.1 plays a role similar to that of the default setting, enhancing training stability even when the step-size warm-up is turned off.

## 9 Conclusions

In this paper, we investigate the convergence behavior of Adafactor on non-convex smooth landscapes, considering bounded stochastic gradients. We introduce a new proxy step-size to decouple the stochastic gradients from the unique adaptive step-size. Additionally, we use new estimations to control the errors introduced by matrix factorization and update clipping in Adafactor.

Our findings reveal that full-batch Adafactor is capable of finding a stationary point, requiring only a step-size $\eta_k \sim \mathcal{O}(1/\sqrt{k})$ and a second moment decay rate $\beta_{2,k} \in (0,1)$, denoting a wide range including the default setup. In the case of stochastic Adafactor without update clipping, the convergence rate can achieve the optimal order $\tilde{\mathcal{O}}(1/\sqrt{T})$ when $\beta_{2,k} = 1 - 1/k^c, c = 1$. However, performance deteriorates as $c$ decreases. This finding is supported by experimental results. We also explore Adafactor with a time-increasing clipping threshold and derive similar convergence results. The empirical results demonstrate that the new clipping threshold provides performance comparable to the default constant setting.

**Limitations** There are several limitations in our work that warrant further investigation. First, the polynomial dependency on $\epsilon_1$ in our convergence bounds may be further improved to a better dependency, such as $\log(1/\epsilon_1)$. Second, although we provide convergence results for several variants of Adafactor and demonstrate comparable performance to the original one in experiments, the convergence bound for stochastic vanilla Adafactor remains unknown. Finally, our experimental results primarily focus on traditional deep learning tasks due to our GPU limitations. It would be beneficial to test the scalability of our theoretical results, e.g., on large language models.

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

 **A   Proof detail for full-batch case**

 We first provide the full-batch Adafactor as follows. The only difference to Algorithm (1) is the
 replacement of stochastic gradient by deterministic gradient $\nabla f(\boldsymbol{X}_k)$ at each iteration.

---

**Algorithm 2** Full-batch Adafactor

---

**Input:** Initialization point $\boldsymbol{X}_1 \in \mathbb{R}^{n \times m}$, $\boldsymbol{R}_0 = \boldsymbol{0}_n, \boldsymbol{C}_0 = \boldsymbol{0}_m^\top$, relative step-sizes $\{\rho_k\}_{k \geq 1}$, decay
rate $\{\beta_{2,k}\}_{k \geq 1} \in [0, 1)$, regularization constants $\epsilon_1, \epsilon_2 > 0$, clipping threshold $d$.
**for** $k = 1, \cdots, T$ **do**
$\quad \bar{\boldsymbol{G}}_k = \nabla f(\boldsymbol{X}_k)$;
$\quad \bar{\boldsymbol{R}}_k = \beta_{2,k} \bar{\boldsymbol{R}}_{k-1} + (1 - \beta_{2,k})(\bar{\boldsymbol{G}}_k \odot \bar{\boldsymbol{G}}_k + \epsilon_1 \boldsymbol{1}_n \boldsymbol{1}_m^\top) \boldsymbol{1}_m$;
$\quad \bar{\boldsymbol{C}}_k = \beta_{2,k} \bar{\boldsymbol{C}}_{k-1} + (1 - \beta_{2,k}) \boldsymbol{1}_n^\top (\bar{\boldsymbol{G}}_k \odot \bar{\boldsymbol{G}}_k + \epsilon_1 \boldsymbol{1}_n \boldsymbol{1}_m^\top)$;
$\quad \bar{\boldsymbol{W}}_k = (\bar{\boldsymbol{R}}_k \bar{\boldsymbol{C}}_k) / \boldsymbol{1}_n^\top \bar{\boldsymbol{R}}_k$;
$\quad \bar{\boldsymbol{U}}_k = \bar{\boldsymbol{G}}_k / \sqrt{\bar{\boldsymbol{W}}_k}$;
$\quad \hat{\eta}_k = \max\{\epsilon_2, \text{RMS}(\boldsymbol{X}_k)\} \rho_k / \max\{1, \text{RMS}(\bar{\boldsymbol{U}}_k)/d\}$;
$\quad \boldsymbol{X}_{k+1} = \boldsymbol{X}_k - \hat{\eta}_k \cdot \bar{\boldsymbol{G}}_k / \sqrt{\bar{\boldsymbol{W}}_k}$;
**end for**

---

 Then, we provide the detailed version of Theorem 5.1 as follows.

 **Theorem A.1.** *Let* $\{\boldsymbol{X}_k\}_{k \geq 1}$ *be generated by Algorithm 2. If Assumptions (A1), (A2) hold,*
 $\|\nabla f(\boldsymbol{X}_k)\|_F \leq G, \forall k \geq 1$ *and*

$$\rho_k = \rho_0/\sqrt{k}, \quad 0 < \beta_{2,k} < 1, \quad \forall k \geq 1,$$

 *for some positive constant* $\rho_0$, *then for any* $T \geq 1$,

$$\min_{k \in [T]} \|\nabla f(\boldsymbol{X}_k)\|_F^2 \leq \frac{A_0 A_1 (f(\boldsymbol{X}_1) - f^* + \Delta_0^2 \log T + \Delta_0^2)}{\sqrt{T}},$$

$$\min_{k \in [T]} \|\nabla f(\boldsymbol{X}_k)\|_F^2 \leq \frac{A_0 A_1' (f(\boldsymbol{X}_1) - f^* + \tilde{\Delta}_0^2 \log T + \Delta_0^2)}{\sqrt{T}}, \tag{9}$$

 *where we define*

$$\Theta_{\min} = \min_{k \in [T]} \|\boldsymbol{X}_k\|_\infty, \quad \Theta_{\max} = \max_{k \in [T]} \|\boldsymbol{X}_k\|_\infty, \quad \mathcal{G} = G^2 + mn\epsilon_1, \tag{10}$$

 *and the other constant parameters are given by*

$$\Delta_0^2 = \frac{Ld^2 mn(\epsilon_2 + \Theta_{\max})^2 \rho_0^2}{2}, \quad \tilde{\Delta}_0^2 = \frac{LG^2 \mathcal{G}(\epsilon_2 + \Theta_{\max})^2 \rho_0^2}{2mn\epsilon_1^2 (1 - \beta_{2,1})^2},$$

$$A_0 = \frac{\max\left\{1, \frac{G\sqrt{\mathcal{G}}}{d\epsilon_1 mn(1 - \beta_{2,1})}\right\}}{\rho_0 \max\{\epsilon_2, \Theta_{\min}\}}, A_1 = \sqrt{G^4 + G^2(m + n)\epsilon_1 + mn\epsilon_1^2}, \tag{11}$$

$$A_1' = \sqrt{2\left(\frac{G^4}{mn\epsilon_1} + G^2 + \epsilon_1\right)}.$$

 **A.1   Preliminary**

 We first denote the auxiliary matrix $\bar{\boldsymbol{G}}_{k,\epsilon_1}^2 = \bar{\boldsymbol{G}}_k \odot \bar{\boldsymbol{G}}_k + \epsilon_1 \boldsymbol{1}_n \boldsymbol{1}_m^\top$. In addition, we define $\bar{\boldsymbol{V}}_k = $
 $\left(\bar{v}_{ij}^{(k)}\right)_{ij}$ as follows,

$$\bar{\boldsymbol{V}}_0 = \boldsymbol{0}_{n \times m}, \quad \bar{\boldsymbol{V}}_k = \beta_{2,k} \bar{\boldsymbol{V}}_{k-1} + (1 - \beta_{2,k}) \bar{\boldsymbol{G}}_{k,\epsilon_1}^2, \quad k \geq 1. \tag{12}$$

 To simplify the notation, we let $\bar{\boldsymbol{G}}_k = \left(\bar{g}_{ij}^{(k)}\right)_{ij}$, $R_{\bar{\boldsymbol{V}}_k}^{(i)}, C_{\bar{\boldsymbol{V}}_k}^{(j)}$ and $S_{\bar{\boldsymbol{V}}_k}$ be the $i$-th row sum, $j$-th column
 sum and the coordinate sum of $\bar{\boldsymbol{V}}_k$ respectively. The same definition principal is applied to the
 notation $R_{\bar{\boldsymbol{G}}_{k,\epsilon_1}^2}^{(i)}$ and $C_{\bar{\boldsymbol{G}}_{k,\epsilon_1}^2}^{(j)}$. We also use $\bar{w}_{ij}^{(k)}, \bar{v}_{ij}^{(k)}, \bar{u}_{ij}^{(k)}$ to denote the coordinates of $\bar{\boldsymbol{W}}_k, \bar{\boldsymbol{V}}_k, \bar{\boldsymbol{U}}_k$
 in Algorithm 2 respectively. We also define values $\mathcal{G}_1, \mathcal{G}_2, \mathcal{G}$ as follows:

$$\mathcal{G}_1 = G^2 + m\epsilon_1, \quad \mathcal{G}_2 = G^2 + n\epsilon_1, \quad \mathcal{G} = G^2 + mn\epsilon_1. \tag{13}$$

### A.2 Technical lemmas

Following the descent lemma for a $L$-smooth objective function $f$, we derive that

$$f(\boldsymbol{Y}) \leq f(\boldsymbol{X}) + \langle \nabla f(\boldsymbol{X}), \boldsymbol{Y} - \boldsymbol{X} \rangle + \frac{L}{2}\|\boldsymbol{Y} - \boldsymbol{X}\|_F^2, \quad \forall \boldsymbol{X}, \boldsymbol{Y} \in \mathbb{R}^{n \times m}. \tag{14}$$

In the following, we will provide some necessary technical lemmas.

**Lemma A.1.** *Let $\beta_{2,k} \in (0,1)$ and $\Gamma_k$ be defined by*

$$\Gamma_0 = 0, \quad \Gamma_k = \beta_{2,k}\Gamma_{k-1} + (1 - \beta_{2,k}), \quad \forall k \geq 1.$$

*Then, $(1 - \beta_{2,1}) \leq \Gamma_k \leq 1, \forall k \geq 1$.*

*Proof.* We could prove the result by induction. Since $\Gamma_0 = 0$, it's easy to derive that $(1 - \beta_{2,1}) = \Gamma_1 \leq 1$. Suppose that for any $j \in [k-1]$, $(1 - \beta_{2,1}) \leq \Gamma_j \leq 1$. Then

$$\Gamma_k \geq \beta_{2,k}(1 - \beta_{2,1}) + (1 - \beta_{2,k}) \geq 1 - \beta_{2,1}, \quad \Gamma_k \leq \beta_{2,k} + (1 - \beta_{2,k}) \leq 1.$$

The induction is then complete. $\qquad\square$

**Lemma A.2.** *Let $\bar{\boldsymbol{V}}_k$ be defined in (12). For any $k \geq 0$, it holds that*

$$\bar{\boldsymbol{R}}_k = \bar{\boldsymbol{V}}_k \mathbf{1}_m, \quad \bar{\boldsymbol{C}}_k = \mathbf{1}_n^\top \bar{\boldsymbol{V}}_k, \quad S_{\bar{\boldsymbol{V}}_k} = \mathbf{1}_n^\top \bar{\boldsymbol{R}}_k = \mathbf{1}_n^\top \bar{\boldsymbol{V}}_k \mathbf{1}_m.$$

*As a consequence,*

$$R_{\bar{\boldsymbol{V}}_k}^{(i)} = \beta_{2,k} R_{\bar{\boldsymbol{V}}_{k-1}}^{(i)} + (1 - \beta_{2,k})R_{\bar{\boldsymbol{G}}_{k,\epsilon_1}^2}^{(i)}, \quad C_{\bar{\boldsymbol{V}}_k}^{(j)} = \beta_{2,k} C_{\bar{\boldsymbol{V}}_{k-1}}^{(j)} + (1 - \beta_{2,k})C_{\bar{\boldsymbol{G}}_{k,\epsilon_1}^2}^{(j)}. \tag{15}$$

*Proof.* Note that $\bar{\boldsymbol{R}}_0 = \bar{\boldsymbol{V}}_0 \mathbf{1}_m = \mathbf{0}_n$ and $\bar{\boldsymbol{C}}_0 = \mathbf{1}_n^\top \bar{\boldsymbol{V}}_0 = \mathbf{0}_m^\top$. Suppose that for any $j \leq k-1$, $\bar{\boldsymbol{R}}_j = \bar{\boldsymbol{V}}_j \mathbf{1}_m, \bar{\boldsymbol{C}}_j = \mathbf{1}_n^\top \bar{\boldsymbol{V}}_j$. Then using the updated rule in Algorithm 2 and (12),

$$\begin{aligned}
\bar{\boldsymbol{R}}_k &= \beta_{2,k}\bar{\boldsymbol{R}}_{k-1} + (1 - \beta_{2,k})\bar{\boldsymbol{G}}_{k,\epsilon_1}^2 \mathbf{1}_m = \left(\beta_{2,k}\bar{\boldsymbol{V}}_{k-1} + (1 - \beta_{2,k})\bar{\boldsymbol{G}}_{k,\epsilon_1}^2\right)\mathbf{1}_m = \bar{\boldsymbol{V}}_k \mathbf{1}_m, \\
\bar{\boldsymbol{C}}_k &= \beta_{2,k}\bar{\boldsymbol{C}}_{k-1} + (1 - \beta_{2,k})\mathbf{1}_n^\top \bar{\boldsymbol{G}}_{k,\epsilon_1}^2 = \mathbf{1}_n^\top \left(\beta_{2,k}\bar{\boldsymbol{V}}_{k-1} + (1 - \beta_{2,k})\bar{\boldsymbol{G}}_{k,\epsilon_1}^2\right) = \mathbf{1}_n^\top \bar{\boldsymbol{V}}_k.
\end{aligned} \tag{16}$$

Since $S_{\bar{\boldsymbol{V}}_k}$ represents the coordinate sum of $\bar{\boldsymbol{V}}_k$, we could derive that

$$S_{\bar{\boldsymbol{V}}_k} = \sum_{i=1}^n \sum_{j=1}^m \bar{v}_{ij}^{(k)} = \mathbf{1}_n^\top \bar{\boldsymbol{R}}_k = \mathbf{1}_n^\top \bar{\boldsymbol{V}}_k \mathbf{1}_m. \tag{17}$$

Since $R_{\bar{\boldsymbol{V}}_k}^{(i)}$ denotes the $i$-th row sum of $\bar{\boldsymbol{V}}_k$, it's the $i$-th coordinate of $\bar{\boldsymbol{R}}_k$. Hence, for each coordinate of $\bar{\boldsymbol{R}}_k$, using (16),

$$R_{\bar{\boldsymbol{V}}_k}^{(i)} = \beta_{2,k} R_{\bar{\boldsymbol{V}}_{k-1}}^{(i)} + (1 - \beta_{2,k})R_{\bar{\boldsymbol{G}}_{k,\epsilon_1}^2}^{(i)}.$$

Similarly, we could derive the results related to $C_{\bar{\boldsymbol{V}}_k}^{(j)}$. $\qquad\square$

**Lemma A.3.** *Following the parameter setting in (3), for any $i \in [n], j \in [m], k \geq 1$, it holds that*

$$R_{\bar{\boldsymbol{V}}_k}^{(i)} \in [m\epsilon_1(1 - \beta_{2,1}), \mathcal{G}_1], \quad C_{\bar{\boldsymbol{V}}_k}^{(j)} \in [n\epsilon_1(1 - \beta_{2,1}), \mathcal{G}_2], \quad S_{\bar{\boldsymbol{V}}_k} \in [mn\epsilon_1(1 - \beta_{2,1}), \mathcal{G}].$$

*Proof.* Recalling the definition of $\bar{\boldsymbol{V}}_k$ in (12) and $\|\nabla f(\boldsymbol{X}_k)\|_F \leq G, \forall k \geq 1$, we derive that

$$\begin{aligned}
S_{\bar{\boldsymbol{V}}_k} &= \sum_{i=1}^n \sum_{j=1}^m \bar{v}_{ij}^{(k)} = \sum_{i=1}^n \sum_{j=1}^m \sum_{p=1}^k (1 - \beta_{2,p})\left(\left(\bar{g}_{ij}^{(p)}\right)^2 + \epsilon_1\right)\left(\prod_{l=p+1}^k \beta_{2,l}\right) \\
&\leq \sum_{p=1}^k (1 - \beta_{2,p})\left(\prod_{l=p+1}^k \beta_{2,l}\right)\|\bar{\boldsymbol{G}}_p\|_F^2 + \Gamma_k mn\epsilon_1 \leq G^2 \Gamma_k + mn\epsilon_1 \leq \mathcal{G},
\end{aligned} \tag{18}$$

where the last inequality comes from Lemma A.1. Following (18) and Lemma A.1, we also derive that

$$S_{\bar{\boldsymbol{V}}_k} \geq mn\epsilon_1\Gamma_k \geq mn\epsilon_1(1 - \beta_{2,1}).$$

We also derive the upper bounds for $R_{\bar{\boldsymbol{V}}_k}^{(i)}$ and $C_{\bar{\boldsymbol{V}}_k}^{(j)}$ as follows,

$$R_{\bar{\boldsymbol{V}}_k}^{(i)} = \sum_{j=1}^{m} \bar{v}_{ij}^{(k)} \leq \sum_{p=1}^{k}(1 - \beta_{2,p})\left(\prod_{l=p+1}^{k} \beta_{2,l}\right)\|\bar{\boldsymbol{G}}_p\|_F^2 + \Gamma_k m\epsilon_1 \leq G^2\Gamma_k + m\epsilon_1 \leq \mathcal{G}_1,$$

$$C_{\bar{\boldsymbol{V}}_k}^{(j)} = \sum_{i=1}^{n} \bar{v}_{ij}^{(k)} \leq \sum_{p=1}^{k}(1 - \beta_{2,p})\left(\prod_{l=p+1}^{k} \beta_{2,l}\right)\|\bar{\boldsymbol{G}}_p\|_F^2 + \Gamma_k n\epsilon_1 \leq G^2\Gamma_k + n\epsilon_1 \leq \mathcal{G}_2. \quad (19)$$

Similarly, the lower bound could be derived by

$$R_{\bar{\boldsymbol{V}}_k}^{(i)} \geq m\epsilon_1\Gamma_k \geq m\epsilon_1(1 - \beta_{2,1}), \quad C_{\bar{\boldsymbol{V}}_k}^{(j)} \geq n\epsilon_1\Gamma_k \geq n\epsilon_1(1 - \beta_{2,1}).$$

$\square$

## A.3   Proof of Theorem A.1

Now we move to prove the main result. Using (14) and the updated rule in Algorithm 2,

$$f(\boldsymbol{X}_{k+1}) \leq f(\boldsymbol{X}_k) + \langle \bar{\boldsymbol{G}}_k, \boldsymbol{X}_{k+1} - \boldsymbol{X}_k\rangle + \frac{L}{2}\|\boldsymbol{X}_{k+1} - \boldsymbol{X}_k\|_F^2$$

$$= f(\boldsymbol{X}_k) - \hat{\eta}_k\left\langle \bar{\boldsymbol{G}}_k, \frac{\bar{\boldsymbol{G}}_k}{\sqrt{\bar{\boldsymbol{W}}_k}}\right\rangle + \frac{L\hat{\eta}_k^2}{2}\left\|\frac{\bar{\boldsymbol{G}}_k}{\sqrt{\bar{\boldsymbol{W}}_k}}\right\|_F^2.$$

We then re-arrange the order, sum up both sides over $k \in [t]$ and apply $f(\boldsymbol{X}_{t+1}) \geq f^*$ from Assumption (A2) to get,

$$\underbrace{\sum_{k=1}^{t} \hat{\eta}_k\left\|\frac{\bar{\boldsymbol{G}}_k}{\sqrt[4]{\bar{\boldsymbol{W}}_k}}\right\|_F^2}_{(\mathbf{a})} \leq f(\boldsymbol{X}_1) - f^* + \underbrace{\frac{L}{2}\sum_{k=1}^{t} \hat{\eta}_k^2\left\|\frac{\bar{\boldsymbol{G}}_k}{\sqrt{\bar{\boldsymbol{W}}_k}}\right\|_F^2}_{(\mathbf{b})}. \quad (20)$$

Since $\Theta_{\min} \leq \|\boldsymbol{X}_k\|_\infty \leq \Theta_{\max}$, we have $\Theta_{\min} \leq \mathrm{RMS}(\boldsymbol{X}_k) \leq \Theta_{\max}$ for any $k \geq 1$. Hence, using $\hat{\eta}_k$ defined in Algorithm 2,

$$\hat{\eta}_k = \frac{\max\{\epsilon_2, \mathrm{RMS}(\boldsymbol{X}_k)\}\rho_k}{\max\left\{1, \|\bar{\boldsymbol{U}}_k\|_F/(d\sqrt{mn})\right\}} \leq (\epsilon_2 + \Theta_{\max})\rho_k \min\left\{1, \frac{d\sqrt{mn}}{\|\bar{\boldsymbol{U}}_k\|_F}\right\}. \quad (21)$$

Using (21), $\bar{\boldsymbol{U}}_k = \bar{\boldsymbol{G}}_k/\sqrt{\bar{\boldsymbol{W}}_k}$, $\Delta_0$ in (11) and $\rho_k = \rho_0/\sqrt{k}$, we thus derive that

$$(\mathbf{b}) \leq \frac{Ld^2mn(\epsilon_2 + \Theta_{\max})^2}{2}\sum_{k=1}^{t} \rho_k^2 \cdot \frac{\|\bar{\boldsymbol{U}}_k\|_F^2}{\|\bar{\boldsymbol{U}}_k\|_F^2} = \Delta_0^2\sum_{k=1}^{t}\frac{1}{k}. \quad (22)$$

To lower bound $(\mathbf{a})$, we first discuss the maximum operator inside $\hat{\eta}_k$. Let

$$E_1 = \left\{k \in [t] \mid \|\bar{\boldsymbol{U}}_k\|_F \geq d\sqrt{mn}\right\}, \quad E_2 = \left\{k \in [t] \mid \|\bar{\boldsymbol{U}}_k\|_F \leq d\sqrt{mn}\right\}.$$

When $k \in E_1$, since $\|\boldsymbol{X}_k\|_\infty \geq \Theta_{\min}$, it derives that

$$\hat{\eta}_k \geq \frac{d\sqrt{mn}\max\{\epsilon_2, \Theta_{\min}\}\rho_k}{\|\bar{\boldsymbol{U}}_k\|_F}. \quad (23)$$

Using Lemma A.2, we first derive that $\bar{w}_{ij}^{(k)} = (R_{\bar{\boldsymbol{V}}_k}^{(i)}C_{\bar{\boldsymbol{V}}_k}^{(j)})/S_{\bar{\boldsymbol{V}}_k}$. Then, applying Lemma A.3 and $\|\nabla f(\boldsymbol{X}_k)\|_F \leq G$, we could upper bound $\|\bar{\boldsymbol{U}}_k\|_F^2$ as follows,

$$\|\bar{\boldsymbol{U}}_k\|_F^2 = \sum_{i=1}^{n}\sum_{j=1}^{m}\frac{\left(\bar{g}_{ij}^{(k)}\right)^2 S_{\bar{\boldsymbol{V}}_k}}{R_{\bar{\boldsymbol{V}}_k}^{(i)}C_{\bar{\boldsymbol{V}}_k}^{(j)}} \leq \frac{\|\bar{\boldsymbol{G}}_k\|_F^2\mathcal{G}}{mn\epsilon_1^2(1 - \beta_{2,1})^2} \leq \frac{G^2\mathcal{G}}{mn\epsilon_1^2(1 - \beta_{2,1})^2}. \quad (24)$$

Hence, combining with (23) and (24), we have

$$\sum_{k \in E_1} \hat{\eta}_k \left\| \frac{\bar{\boldsymbol{G}}_k}{\sqrt[4]{\bar{\boldsymbol{W}}_k}} \right\|_F^2 \geq d\sqrt{mn} \max\{\epsilon_2, \Theta_{\min}\} \sum_{k \in E_1} \frac{\rho_k}{\|\bar{\boldsymbol{U}}_k\|_F} \left\| \frac{\bar{\boldsymbol{G}}_k}{\sqrt[4]{\bar{\boldsymbol{W}}_k}} \right\|_F^2$$

$$\geq \frac{d\epsilon_1 mn(1 - \beta_{2,1}) \max\{\epsilon_2, \Theta_{\min}\}}{G\sqrt{\mathcal{G}}} \sum_{k \in E_1} \rho_k \left\| \frac{\bar{\boldsymbol{G}}_k}{\sqrt[4]{\bar{\boldsymbol{W}}_k}} \right\|_F^2. \qquad (25)$$

When $k \in E_2$, we obtain that $\hat{\eta}_k = \max\{\epsilon_2, \mathrm{RMS}(\boldsymbol{X}_k)\}\rho_k \geq \max\{\epsilon_2, \Theta_{\min}\}\rho_k$ and thus

$$\sum_{k \in E_2} \hat{\eta}_k \left\| \frac{\bar{\boldsymbol{G}}_k}{\sqrt[4]{\bar{\boldsymbol{W}}_k}} \right\|_F^2 \geq \max\{\epsilon_2, \Theta_{\min}\} \sum_{k \in E_2} \rho_k \left\| \frac{\bar{\boldsymbol{G}}_k}{\sqrt[4]{\bar{\boldsymbol{W}}_k}} \right\|_F^2. \qquad (26)$$

Combining with (25) and (26), we derive that

$$(\mathbf{a}) \geq \max\{\epsilon_2, \Theta_{\min}\} \min \left\{ 1, \frac{d\epsilon_1 mn(1 - \beta_{2,1})}{G\sqrt{\mathcal{G}}} \right\} \sum_{k=1}^t \rho_k \left\| \frac{\bar{\boldsymbol{G}}_k}{\sqrt[4]{\bar{\boldsymbol{W}}_k}} \right\|_F^2. \qquad (27)$$

We also derive from Lemma A.2 and Lemma A.3 that for any $i \in [n], j \in [m]$,

$$\bar{w}_{ij}^{(k)} = \frac{R_{\bar{\boldsymbol{V}}_k}^{(i)} C_{\bar{\boldsymbol{V}}_k}^{(j)}}{S_{\bar{\boldsymbol{V}}_k}} \leq \frac{R_{\bar{\boldsymbol{V}}_k}^{(i)} C_{\bar{\boldsymbol{V}}_k}^{(j)}}{\sqrt{R_{\bar{\boldsymbol{V}}_k}^{(i)} C_{\bar{\boldsymbol{V}}_k}^{(j)}}} \leq \sqrt{R_{\bar{\boldsymbol{V}}_k}^{(i)} C_{\bar{\boldsymbol{V}}_k}^{(j)}} \leq \sqrt{\mathcal{G}_1 \mathcal{G}_2}. \qquad (28)$$

Using (28), we have

$$\left\| \frac{\bar{\boldsymbol{G}}_k}{\sqrt[4]{\bar{\boldsymbol{W}}_k}} \right\|_F^2 = \sum_{i=1}^n \sum_{j=1}^m \frac{\left(\bar{g}_{ij}^{(k)}\right)^2}{\sqrt{\bar{w}_{ij}^{(k)}}} \geq \frac{\|\bar{\boldsymbol{G}}_k\|_F^2}{\sqrt{\mathcal{G}_1 \mathcal{G}_2}} = \frac{\|\bar{\boldsymbol{G}}_k\|_F^2}{A_1}, \qquad (29)$$

where $A_1$ has been defined in (11). Plugging (29) into (27), we derive that

$$(\mathbf{a}) \geq \frac{\max\{\epsilon_2, \Theta_{\min}\}}{A_1} \min \left\{ 1, \frac{d\epsilon_1 mn(1 - \beta_{2,1})}{G\sqrt{\mathcal{G}}} \right\} \sum_{k=1}^t \rho_k \|\bar{\boldsymbol{G}}_k\|_F^2. \qquad (30)$$

Plugging (22) and (30) into (20), and using $\rho_k = \rho_0/\sqrt{k}$, we thus derive that

$$\min_{k \in [t]} \|\bar{\boldsymbol{G}}_k\|_F^2 \sum_{k=1}^t \frac{1}{\sqrt{k}} \leq \sum_{k=1}^t \frac{\rho_k \|\bar{\boldsymbol{G}}_k\|_F^2}{\rho_0} \leq A_0 A_1 \left( f(\boldsymbol{X}_1) - f^* + \Delta_0^2 \sum_{k=1}^t \frac{1}{k} \right),$$

where $A_0$ is given in (11). Moreover, we have the following results,

$$\sum_{k=1}^t \frac{1}{k} \leq 1 + \int_1^t \frac{1}{x} dx = 1 + \log t, \quad \sum_{k=1}^t \frac{1}{\sqrt{k}} \geq \sqrt{t}. \qquad (31)$$

We thus derive the first desired result in (9) as follows,

$$\min_{k \in [t]} \|\bar{\boldsymbol{G}}_k\|_F^2 \leq \frac{A_0 A_1}{\sqrt{t}} \left( f(\boldsymbol{X}_1) - f^* + \Delta_0^2 + \Delta_0^2 \log t \right). \qquad (32)$$

**Avoiding the curse of dimension** To derive a free-dimension numerator bound, we first derive from (21) and (24) with $\rho_k = \rho_0/\sqrt{k}$ that

$$(\mathbf{b}) \leq \frac{L(\epsilon_2 + \Theta_{\max})^2}{2} \sum_{k=1}^t \rho_k^2 \|\bar{\boldsymbol{U}}_k\|_F^2 \leq \frac{LG^2 \mathcal{G}(\epsilon_2 + \Theta_{\max})^2}{2mn\epsilon_1^2(1 - \beta_{2,1})^2} \sum_{k=1}^t \rho_k^2 = \tilde{\Delta}_0^2 \sum_{k=1}^t \frac{1}{k}, \qquad (33)$$

where $\tilde{\Delta}_0$ has been defined in (11). In addition, we derive from Lemma A.2, Lemma A.3 and (13) that

$$\bar{w}_{ij}^{(k)} = \frac{R_{\bar{\boldsymbol{V}}_k}^{(i)} C_{\bar{\boldsymbol{V}}_k}^{(j)}}{S_{\bar{\boldsymbol{V}}_k}} \leq \frac{2\mathcal{G}_1 \mathcal{G}_2}{mn\epsilon_1} \leq 2 \left( \frac{G^4}{mn\epsilon_1} + G^2 + \epsilon_1 \right) = (A_1')^2, \qquad (34)$$

where we use $m + n \leq mn$ and $A_1'$ in (11). Thereby, we have

$$\left\| \frac{\bar{\boldsymbol{G}}_k}{\sqrt[4]{\bar{\boldsymbol{W}}_k}} \right\|_F^2 = \sum_{i=1}^n \sum_{j=1}^m \frac{\left(\bar{g}_{ij}^{(k)}\right)^2}{\sqrt{\bar{w}_{ij}^{(k)}}} \geq \frac{\|\bar{\boldsymbol{G}}_k\|_F^2}{A_1'}.$$

Combining with (27), we thus derive that

$$(\mathbf{a}) \geq \frac{\max\{\epsilon_2, \Theta_{\min}\}}{A_1'} \min\left\{1, \frac{d\epsilon_1 mn(1 - \beta_{2,1})}{G\sqrt{\mathcal{G}}}\right\} \sum_{k=1}^t \rho_k \|\bar{\boldsymbol{G}}_k\|_F^2 \tag{35}$$

Plugging (33) and (35) into (20), and using $\rho_k = \rho_0/\sqrt{k}$, we derive that

$$\min_{k\in[t]} \|\bar{\boldsymbol{G}}_k\|_F^2 \sum_{k=1}^t \frac{1}{\sqrt{k}} \leq \sum_{k=1}^t \frac{\rho_k \|\bar{\boldsymbol{G}}_k\|_F^2}{\rho_0} \leq A_0 A_1' \left(f(\boldsymbol{X}_1) - f^* + \tilde{\Delta}_0^2 \sum_{k=1}^t \frac{1}{k}\right),$$

where $A_0$ has been defined in (11). Using (31), we derive the second desired result in (9).

$$\min_{k\in[t]} \|\bar{\boldsymbol{G}}_k\|_F^2 \leq \frac{A_0 A_1'}{\sqrt{t}} \left(f(\boldsymbol{X}_1) - f^* + \tilde{\Delta}_0^2 + \tilde{\Delta}_0^2 \log t\right). \tag{36}$$

# B   Proof detail for stochastic Adafactor without update clipping

We first provide the detailed version of Theorem 6.1.

**Theorem B.1** (*Formal statement of Theorem 6.1*). *Let $\{\boldsymbol{X}_k\}_{k\geq 1}$ be generated by Algorithm 1 without update clipping where $\eta_k$ is given by (5) for each $k \geq 1$. If Assumptions (A1)-(A4) hold, and*

$$\beta_{2,1} = 1/2, \quad \rho_1 = \rho_0,$$
$$\beta_{2,k} = 1 - 1/k^c, \quad \rho_k = \rho_0/\sqrt{k}, \quad \forall k \geq 2,$$

*for some constants $1/2 \leq c \leq 1, \rho_0 > 0$, then for any $T \geq 1, \delta \in (0, 1)$, we have the following results.*
*When $c = 1$, with probability at least $1 - \delta$,*

$$\min_{k\in[T]} \|\bar{\boldsymbol{G}}_k\|_F^2 \leq \frac{C_0}{\sqrt{T}} \left(C_1 \log\left(\frac{T}{\delta}\right) + C_2 \log T + C_2 + C_3\right), \tag{37}$$

$$\min_{k\in[T]} \|\bar{\boldsymbol{G}}_k\|_F^2 \leq \frac{C_0'}{\sqrt{T}} \left(C_1 \log\left(\frac{T}{\delta}\right) + (C_2' + C_3') \log T + C_2' + C_3'\right). \tag{38}$$

*When $1/2 \leq c < 1$, with probability at least $1 - \delta$,*

$$\min_{k\in[T]} \|\bar{\boldsymbol{G}}_k\|_F^2 \leq \frac{C_0}{\sqrt{T}} \left(C_1 \log\left(\frac{T}{\delta}\right) + \frac{C_2}{1-c} \cdot T^{1-c} + C_2 + C_3\right), \tag{39}$$

$$\min_{k\in[T]} \|\bar{\boldsymbol{G}}_k\|_F^2 \leq \frac{C_0}{\sqrt{T}} \left(C_1 \log\left(\frac{T}{\delta}\right) + \frac{2C_2'}{1-c} \cdot T^{1-c} + C_3' \log T + C_2' + C_3'\right). \tag{40}$$

*Here, $\Theta_{\min}, \Theta_{\max}$ and $\mathcal{G}$ are as in (10), and*

$$C_1 = f(\boldsymbol{X}_1) - f^* + \frac{24G^2(\epsilon_2 + \Theta_{\max})\rho_0}{\sqrt{\epsilon_1}}. \tag{41}$$

*The $C_0, C_2, C_3$ are constants defined as*

$$C_0 = \frac{2\sqrt{2\mathcal{G}}}{\rho_0 \max\{\epsilon_2, \Theta_{\min}\}}, \quad C_3 = \frac{C_2}{4} \log\left(2 + \frac{2G^2}{\epsilon_1}\right),$$

$$C_2 = \frac{32mn\mathcal{G}^{\frac{3}{2}}(\epsilon_2 + \Theta_{\max})\rho_0}{\max\{m, n\}\epsilon_1} + \frac{4Lmn\mathcal{G}(\epsilon_2 + \Theta_{\max})^2\rho_0^2}{\max\{m, n\}\epsilon_1}. \tag{42}$$

The $C_0', C_2', C_3'$ are positive constants (that could be further upper bounded by constants independent from $m, n$), defined by

$$C_0' = \frac{2\sqrt{2\left(\frac{G^2}{mn\epsilon_1} + G + \epsilon_1\right)}}{\rho_0 \max\{\epsilon_2, \Theta_{\min}\}}, C_2' = 4G_3(G_1 + G_2)(\epsilon_2 + \Theta_{\max})\rho_0, C_3' = \frac{LG_3(\epsilon_2 + \Theta_{\max})^2\rho_0^2}{2}, \tag{43}$$

and $G_1, G_2, G_3$ are given by

$$G_1 = \sqrt{6\left(\frac{G^4}{mn\epsilon_1} + G^2 + \epsilon_1\right)}, \quad G_3 = \frac{4(G^4 + G^2 mn\epsilon_1)}{mn\epsilon_1^2},$$

$$G_2 = 2\left(\frac{G^3}{mn\epsilon_1} + \frac{2G^2}{\sqrt{mn\epsilon_1}} + \frac{G}{\sqrt{mn}} + G + \sqrt{\epsilon_1}\right). \tag{44}$$

**Calculation of hyper-parameter dependency** To derive a free dimension bound, we shall use the convergence bounds in (38) and (40). From (43), it's easy to show that $m, n$ could only exist in the denominator of $C_0', C_2', C_3'$, which could avoid the curse of dimension.

To calculate the dependency of $\epsilon_1$, we first show that its dependency in coefficients $C_0, C_1, C_2, C_3$ as follows, based on the assumption that $0 < \epsilon_1 < 1$,

$$C_0 \sim \mathcal{O}(1), \quad C_1 \sim \mathcal{O}(1/\sqrt{\epsilon_1}), \quad C_2 \sim \mathcal{O}(1/\epsilon_1), \quad C_3 \sim \mathcal{O}(C_2 \log(1/\epsilon_1)). \tag{45}$$

Thereby, with the convergence bounds in (37) and (39), it's easy to show that

$$\min_{k \in [T]} \|\bar{\boldsymbol{G}}_k\|_F^2 \le \mathcal{O}\left(\epsilon_1^{-1} \log(1/\epsilon_1)\right). \tag{46}$$

**Proposition B.1.** *Following the same assumptions and settings in Theorem 6.1, then with probability at least $1 - \delta$,*

$$\min_{k \in [T]} \|\bar{\boldsymbol{G}}_k\|_F^2 \le \frac{C_0}{\sqrt{T}}\left(C_1 \log\left(\frac{T}{\delta}\right) + C_2 \sum_{k=1}^{T} \frac{1}{k^c} + C_3\right),$$

*and with probability at least $1 - \delta$,*

$$\min_{k \in [T]} \|\bar{\boldsymbol{G}}_k\|_F^2 \le \frac{C_0'}{\sqrt{T}}\left(C_1 \log\left(\frac{T}{\delta}\right) + C_2' \sum_{k=1}^{T} \frac{1}{k^{c/2+1/2}} + C_3' \sum_{k=1}^{T} \frac{1}{k}\right),$$

*where all constants are given as in Theorem B.1.*

## B.1 Preliminary

We first follow the notations of $\bar{\boldsymbol{G}}_k = \left(\bar{g}_{ij}^{(k)}\right)_{ij}$ and $\mathcal{G}, \mathcal{G}_1, \mathcal{G}_2$ in (13). Let $\boldsymbol{G}_k = \left(g_{ij}^{(k)}\right)_{ij}$ and $\boldsymbol{\xi}_k = \boldsymbol{G}_k - \bar{\boldsymbol{G}}_k$. We also define $\boldsymbol{G}_{k,\epsilon_1}^2 = \boldsymbol{G}_k \odot \boldsymbol{G}_k + \epsilon_1 \mathbf{1}_n \mathbf{1}_m^\top$ and $\boldsymbol{V}_k = \left(v_{ij}^{(k)}\right)_{ij}$ as follows,

$$\boldsymbol{V}_0 = \mathbf{0}_{n \times m}, \quad \boldsymbol{V}_k = \beta_{2,k}\boldsymbol{V}_{k-1} + (1 - \beta_{2,k})\boldsymbol{G}_{k,\epsilon_1}^2, \quad k \ge 1. \tag{47}$$

We also define $R_{\boldsymbol{V}_k}^{(i)}, C_{\boldsymbol{V}_k}^{(j)}$ and $S_{\boldsymbol{V}_k}$ as the $i$-th row sum, $j$-th column sum and coordinate sum of $\boldsymbol{V}_k$ respectively. $R_{\boldsymbol{G}_{k,\epsilon_1}^2}^{(i)}$ and $C_{\boldsymbol{G}_{k,\epsilon_1}^2}^{(j)}$ represent the same definitions with respect to $\boldsymbol{G}_{k,\epsilon_1}^2$. Then, using a similar deduction in Lemma A.2, we also obtain that for all $k \ge 1$,

$$R_{\boldsymbol{V}_k}^{(i)} = \beta_{2,k}R_{\boldsymbol{V}_{k-1}}^{(i)} + (1 - \beta_{2,k})\boldsymbol{G}_{k,\epsilon_1}^2 \mathbf{1}_m, \quad C_{\boldsymbol{V}_k}^{(j)} = \beta_{2,k}C_{\boldsymbol{V}_{k-1}}^{(j)} + (1 - \beta_{2,k})\mathbf{1}_n^\top \boldsymbol{G}_{k,\epsilon_1}^2. \tag{48}$$

As a consequence of (48), each coordinate of $\boldsymbol{W}_k$ satisfies that

$$w_{ij}^{(k)} = \frac{R_{\boldsymbol{V}_k}^{(i)}C_{\boldsymbol{V}_k}^{(j)}}{S_{\boldsymbol{V}_k}} = \frac{\left(\beta_{2,k}R_{\boldsymbol{V}_{k-1}}^{(i)} + (1 - \beta_{2,k})R_{\boldsymbol{G}_{k,\epsilon_1}^2}^{(i)}\right)\left(\beta_{2,k}C_{\boldsymbol{V}_{k-1}}^{(j)} + (1 - \beta_{2,k})C_{\boldsymbol{G}_{k,\epsilon_1}^2}^{(j)}\right)}{\beta_{2,k}S_{\boldsymbol{V}_{k-1}} + (1 - \beta_{2,k})S_{\boldsymbol{G}_{k,\epsilon_1}^2}}. \tag{49}$$

531 Next, we introduce a proxy step-size matrix $\boldsymbol{A}_k = \left( a_{ij}^{(k)} \right)_{ij}$ such that

$$a_{ij}^{(k)} = \frac{\left( \beta_{2,k} R_{\boldsymbol{V}_{k-1}}^{(i)} + (1 - \beta_{2,k})\mathcal{G}_1 \right) \left( \beta_{2,k} C_{\boldsymbol{V}_{k-1}}^{(j)} + (1 - \beta_{2,k})\mathcal{G}_2 \right)}{\beta_{2,k} S_{\boldsymbol{V}_{k-1}} + (1 - \beta_{2,k})\mathcal{G}}. \tag{50}$$

532 The proxy step-size technique is a standard way in the convergence analysis of adaptive methods,
533 e.g., [32, 8]. We provide a new proxy step-size in (50) to handle the matrix factorization in Adafactor.
534 This construction satisfies two properties. First, it's independent from $\boldsymbol{Z}_k$ in order to disrupt the
535 correlation of stochastic gradients and adaptive step-sizes. Second, it needs to remain sufficiently
536 close to the original adaptive step-size $w_{ij}^{(k)}$ to avoid generating divergent terms.

## B.2 Technical lemmas

538 In the following, we first provide some more necessary technical lemmas. We introduce a concentra-
539 tion inequality for the martingale difference sequence, see [20] for a proof.

540 **Lemma B.1.** *Suppose that* $\{Z_s\}_{s \in [T]}$ *is a martingale difference sequence with respect to* $\zeta_1, \cdots, \zeta_T$.
541 *Assume that for each* $s \in [T]$, $\sigma_s$ *is a random variable dependent on* $\zeta_1, \cdots, \zeta_{s-1}$ *and satisfies that*

$$\mathbb{E}\left[ \exp\left( \frac{Z_s^2}{\sigma_s^2} \right) \mid \zeta_1, \cdots, \zeta_{s-1} \right] \le \mathrm{e}.$$

542 *Then for any* $\lambda > 0$, *and for any* $\delta \in (0, 1)$, *it holds that*

$$\mathbb{P}\left( \sum_{s=1}^{T} Z_s > \frac{1}{\lambda} \log\left( \frac{1}{\delta} \right) + \frac{3}{4}\lambda \sum_{s=1}^{T} \sigma_s^2 \right) \le \delta.$$

543 **Lemma B.2.** *Following the parameter setting in (6), for any* $i \in [n], j \in [m], k \ge 1$, *it holds that*

$$R_{\boldsymbol{G}_{k,\epsilon_1}^2}^{(i)}, R_{\boldsymbol{V}_k}^{(i)} \in [m\epsilon_1/2, \mathcal{G}_1], \quad C_{\boldsymbol{G}_{k,\epsilon_1}^2}^{(j)}, C_{\boldsymbol{V}_k}^{(j)} \in [n\epsilon_1/2, \mathcal{G}_2], \quad S_{\boldsymbol{G}_{k,\epsilon_1}^2}, S_{\boldsymbol{V}_k} \in [mn\epsilon_1/2, \mathcal{G}].$$

544 *Proof.* First, using Assumption (A4), we derive that

$$mn\epsilon_1/2 \le S_{\boldsymbol{G}_{k,\epsilon_1}^2} = \sum_{i=1}^{n} \sum_{j=1}^{m} \left( \left( g_{ij}^{(k)} \right)^2 + \epsilon_1 \right) = \|\boldsymbol{G}_k\|_F^2 + mn\epsilon_1 \le \mathcal{G},$$

$$m\epsilon_1/2 \le R_{\boldsymbol{G}_{k,\epsilon_1}^2}^{(i)} = \sum_{j=1}^{m} \left( \left( g_{ij}^{(k)} \right)^2 + \epsilon_1 \right) \le \|\boldsymbol{G}_k\|_F^2 + m\epsilon_1 \le \mathcal{G}_1,$$

$$n\epsilon_1/2 \le C_{\boldsymbol{G}_{k,\epsilon_1}^2}^{(j)} = \sum_{i=1}^{n} \left( \left( g_{ij}^{(k)} \right)^2 + \epsilon_1 \right) \le \|\boldsymbol{G}_k\|_F^2 + n\epsilon_1 \le \mathcal{G}_2.$$

545 Using the similar deduction for Lemma A.3, we could show that $m\epsilon_1(1 - \beta_{2,1}) \le R_{\boldsymbol{V}_k}^{(i)} \le \mathcal{G}_1$. Since
546 $\beta_{2,1} = 1/2$ from (6), we then obtain the desired result. The bounds for $C_{\boldsymbol{V}_k}^{(j)}, S_{\boldsymbol{V}_k}$ could be also
547 derived by using similar arguments. $\qquad \square$

548 We have the following lemma to upper bound each coordinate of the proxy step-size matrix $\boldsymbol{A}_k$
549 defined in (50) .

550 **Lemma B.3.** *For any* $k \ge 1$, *it holds that*

$$\beta_{2,k}(1 - \beta_{2,k})\epsilon_1 \le a_{ij}^{(k)} \le 2\min\left\{ \mathcal{G}, \frac{G^2}{mn\epsilon_1} + G + \epsilon_1 \right\}, \quad \forall i \in [n], j \in [m].$$

551 *Proof.* We first have

$$\frac{\beta_{2,k} R_{\boldsymbol{V}_{k-1}}^{(i)} + (1 - \beta_{2,k})\mathcal{G}_1}{\beta_{2,k} S_{\boldsymbol{V}_{k-1}} + (1 - \beta_{2,k})\mathcal{G}} \le \frac{\beta_{2,k} R_{\boldsymbol{V}_{k-1}}^{(i)}}{\beta_{2,k} S_{\boldsymbol{V}_{k-1}}} + \frac{(1 - \beta_{2,k})\mathcal{G}_1}{(1 - \beta_{2,k})\mathcal{G}} \le 2. \tag{51}$$

Then, recalling the definition of $a_{ij}^{(k)}$ in (50) and Lemma B.2, it derives that $C_{\boldsymbol{V}_{k-1}}^{(j)} \leq \mathcal{G}_2$ and thereby $\beta_{2,k} C_{\boldsymbol{V}_{k-1}}^{(j)} + (1 - \beta_{2,k})\mathcal{G}_2 \leq \mathcal{G}_2 \leq \mathcal{G}$. Then combining with (51), we derive $a_{ij}^{(k)} \leq 2\mathcal{G}$. We also derive a free dimension bound from Lemma B.2 for $a_{ij}^{(k)}$ as follows,

$$a_{ij}^{(k)} \leq \frac{2\mathcal{G}_1 \mathcal{G}_2}{mn\epsilon_1} = \frac{2(G^2 + G(m+n)\epsilon_1 + mn\epsilon_1^2)}{mn\epsilon_1} \leq 2\left(\frac{G^2}{mn\epsilon_1} + G + \epsilon_1\right),$$

where we use $m + n \leq mn$ when $m, n \geq 2$ and $\beta_{2,k} S_{\boldsymbol{V}_{k-1}} + (1 - \beta_{2,k})\mathcal{G} \geq mn\epsilon_1/2$. To lower bound $a_{ij}^{(k)}$, we derive from Lemma B.2 that $\beta_{2,k} S_{\boldsymbol{V}_{k-1}} + (1 - \beta_{2,k})\mathcal{G} \leq \mathcal{G}$. Thereby,

$$a_{ij}^{(k)} \geq \frac{\beta_{2,k}(1 - \beta_{2,k})\left(R_{\boldsymbol{V}_{k-1}}^{(i)} \mathcal{G}_2 + C_{\boldsymbol{V}_{k-1}}^{(j)} \mathcal{G}_1\right)}{\mathcal{G}} \geq \beta_{2,k}(1 - \beta_{2,k}) \cdot \frac{(m\mathcal{G}_2 + n\mathcal{G}_1)\epsilon_1}{2\mathcal{G}}$$

$$= \beta_{2,k}(1 - \beta_{2,k}) \cdot \frac{[(m+n)G^2 + 2mn\epsilon_1]\epsilon_1}{2(G^2 + mn\epsilon_1)} \geq \beta_{2,k}(1 - \beta_{2,k})\epsilon_1.$$

□

**Lemma B.4.** *Let $\boldsymbol{W}_k$ and $\boldsymbol{V}_k$ be defined in Algorithm 1 without update clipping where $\eta_k$ is given by (5) and (47) respectively. For any $k \geq 1$, it holds that*

$$\left\|\frac{\boldsymbol{G}_k}{\sqrt{\boldsymbol{W}_k}}\right\|_F^2 \leq \frac{2\mathcal{G}}{\max\{m,n\}\epsilon_1}\left\|\frac{\boldsymbol{G}_k}{\sqrt{\boldsymbol{V}_k}}\right\|_F^2.$$

*Proof.* Recalling (49), $v_{ij}^{(k)} \leq R_{\boldsymbol{V}_k}^{(i)}$, $v_{ij}^{(k)} \leq C_{\boldsymbol{V}_k}^{(j)}$ and Lemma B.2, one could verify that

$$\frac{\left(g_{ij}^{(k)}\right)^2}{w_{ij}^{(k)}} = \frac{\left(g_{ij}^{(k)}\right)^2 S_{\boldsymbol{V}_k}}{R_{\boldsymbol{V}_k}^{(i)} C_{\boldsymbol{V}_k}^{(j)}} \leq \frac{2\left(g_{ij}^{(k)}\right)^2 \mathcal{G}}{n\epsilon_1 v_{ij}^{(k)}}, \quad \frac{\left(g_{ij}^{(k)}\right)^2}{w_{ij}^{(k)}} = \frac{\left(g_{ij}^{(k)}\right)^2 S_{\boldsymbol{V}_k}}{R_{\boldsymbol{V}_k}^{(i)} C_{\boldsymbol{V}_k}^{(j)}} \leq \frac{2\left(g_{ij}^{(k)}\right)^2 \mathcal{G}}{m\epsilon_1 v_{ij}^{(k)}},$$

which leads to the desired result that

$$\|\boldsymbol{U}_k\|_F^2 = \left\|\frac{\boldsymbol{G}_k}{\sqrt{\boldsymbol{W}_k}}\right\|_F^2 \leq \frac{2\mathcal{G}}{\max\{m,n\}\epsilon_1}\left\|\frac{\boldsymbol{G}_k}{\sqrt{\boldsymbol{V}_k}}\right\|_F^2.$$

□

The following lemma is inspired by [8, Lemma 5.2] where they considered a constant $\beta_{2,k}$. Here, we generalize the result to the case of time-varying $\beta_{2,k}$ and provide the proof detail.

**Lemma B.5.** *For any $t \geq 1$, if $\beta_{2,k}$ are as in (6), then it holds that*

$$\sum_{k=1}^t (1 - \beta_{2,k})\left\|\frac{\boldsymbol{G}_k}{\sqrt{\boldsymbol{V}_k}}\right\|_F^2 \leq mn \log\left(\frac{2(G^2 + \epsilon_1)}{\epsilon_1}\right) + 4mn \sum_{k=1}^t (1 - \beta_{2,k}).$$

*Proof.* Recalling the definition of $\boldsymbol{V}_k$ and since $\boldsymbol{V}_0 = \boldsymbol{0}_{n \times m}$, we have that for any $k \geq 1$,

$$v_{ij}^{(k)} = \beta_{2,k} v_{ij}^{(k-1)} + (1 - \beta_{2,k})\left[\left(g_{ij}^{(k)}\right)^2 + \epsilon_1\right]$$

$$= \sum_{p=1}^k (1 - \beta_{2,p})\left[\left(g_{ij}^{(p)}\right)^2 + \epsilon_1\right]\left(\prod_{l=p+1}^k \beta_{2,l}\right).$$

Then, we have

$$(1 - \beta_{2,k}) \cdot \frac{\left(g_{ij}^{(k)}\right)^2}{v_{ij}^{(k)}} = \frac{x_k}{y_k + \theta_k}, \tag{52}$$

where we set $y_0 = 0, \theta_0 = 0$ and

$$x_k = (1 - \beta_{2,k})\left(g_{ij}^{(k)}\right)^2, \quad y_k = \sum_{p=1}^{k}(1 - \beta_{2,p})\left(g_{ij}^{(p)}\right)^2\left(\prod_{l=p+1}^{k}\beta_{2,l}\right),$$

$$\theta_k = \epsilon_1\sum_{p=1}^{k}(1 - \beta_{2,p})\left(\prod_{l=p+1}^{k}\beta_{2,l}\right), \quad \forall k \geq 1.$$

Then we have $y_k - x_k = \beta_{2,k}y_{k-1}, \forall k \geq 1$. Moreover, since $y_k \geq x_k$, we could use $\log x \geq 1 - 1/x, \forall x \geq 1$ to derive that

$$\frac{x_k}{y_k + \theta_k} \leq \log(y_k + \theta_k) - \log(y_k + \theta_k - x_k) = \log(y_k + \theta_k) - \log(\beta_{2,k}y_{k-1} + \theta_k)$$

$$= \log\left(\frac{y_k + \theta_k}{y_{k-1} + \theta_{k-1}}\right) + \log\left(\frac{y_{k-1} + \theta_{k-1}}{\beta_{2,k}y_{k-1} + \theta_k}\right).$$

Noting that $\theta_k = \beta_{2,k}\theta_{k-1} + (1 - \beta_{2,k})\epsilon_1$, which leads to $\beta_{2,k}\theta_{k-1} \leq \theta_k$. Hence, we further have

$$\frac{x_k}{y_k + \theta_k} \leq \log\left(\frac{y_k + \theta_k}{y_{k-1} + \theta_{k-1}}\right) + \log\left(\frac{y_{k-1} + \theta_{k-1}}{\beta_{2,k}(y_{k-1} + \theta_{k-1})}\right) = \log\left(\frac{y_k + \theta_k}{y_{k-1} + \theta_{k-1}}\right) - \log\beta_{2,k}.$$
(53)

Hence, summing up on both sides of (52) and (53) over $k \in [t]$, and noting that $x_1 = y_1$, we obtain that

$$\sum_{k=1}^{t}(1 - \beta_{2,k})\cdot\frac{\left(g_{ij}^{(k)}\right)^2}{v_{ij}^{(k)}} = \frac{x_1}{y_1 + \theta_1} + \sum_{k=2}^{t}\frac{x_k}{y_k + \epsilon_k}$$

$$\leq 1 + \log\left(\frac{y_t + \theta_t}{y_1 + \theta_1}\right) - \sum_{k=2}^{t}\log\beta_{2,k}.$$
(54)

Note that $y_1 + \theta_1 \geq (1 - \beta_{2,1})\epsilon_1 = \epsilon_1/2$. Moreover, using Lemma A.1 and Assumption (A4), we have $\theta_t = \Gamma_t\epsilon_1 \leq \epsilon_1$ and $y_t \leq \Gamma_t G^2 \leq G^2$. We then derive that

$$\frac{y_t + \theta_t}{y_1 + \theta_1} \leq \frac{2(G^2 + \epsilon_1)}{\epsilon_1}.$$
(55)

Noting that for $k \geq 2, c \in [1/2, 1], \beta_{2,k} \geq \beta_{2,2} = 1 - 1/2^c \geq 1 - 1/\sqrt{2}$, we then derive that

$$-\log\beta_{2,k} \leq \frac{1 - \beta_{2,k}}{\beta_{2,k}} \leq \frac{\sqrt{2}(1 - \beta_{2,k})}{\sqrt{2} - 1} \leq 4(1 - \beta_{2,k}).$$
(56)

Finally, plugging (55), (56) into (54), and then summing (54) up over $i \in [n], j \in [m]$, we obtain the desired result. $\qquad\square$

Next, we have the following probabilistic result relying on the property of the martingale difference sequence which is commonly used in the analysis of adaptive methods.

**Lemma B.6.** *Following the parameter setting in (6), for any $T \geq 1$ and $\lambda > 0$, with probability at least $1 - \delta, \forall t \in [T]$,*

$$-\sum_{k=1}^{t}\eta_k\left\langle\bar{\boldsymbol{G}}_k, \frac{\boldsymbol{\xi}_k}{\sqrt{\boldsymbol{A}_k}}\right\rangle \leq \frac{1}{4}\sum_{k=1}^{t}\eta_k\left\|\frac{\bar{\boldsymbol{G}}_k}{\sqrt[4]{\boldsymbol{A}_k}}\right\|_F^2 + \frac{24G^2(\epsilon_2 + \Theta_{\max})\rho_0}{\sqrt{\epsilon_1}}\log\left(\frac{T}{\delta}\right).$$

*Proof.* Let $\zeta_k = -\eta_k\left\langle\bar{\boldsymbol{G}}_k, \frac{\boldsymbol{\xi}_k}{\sqrt{\boldsymbol{A}_k}}\right\rangle$ and the filtration $\mathcal{F}_k = \sigma\left(\boldsymbol{Z}_1, \cdots, \boldsymbol{Z}_k\right)$ where $\sigma(\cdot)$ denotes the $\sigma$-algebra. Note that $\eta_k, \bar{\boldsymbol{G}}_k$ and $\boldsymbol{A}_k$ are dependent by $\{\boldsymbol{X}_1, \cdots, \boldsymbol{X}_{k-1}\}$ and thereby $\mathcal{F}_{k-1}$. Since $\boldsymbol{\xi}_k$ is dependent by $\mathcal{F}_k$, we could prove that $\{\zeta_k\}_{k\geq 1}$ is a martingale difference sequence since

$$\mathbb{E}\left[\zeta_k \mid \mathcal{F}_{k-1}\right] = -\eta_k\left\langle\bar{\boldsymbol{G}}_k, \frac{\mathbb{E}\left[\boldsymbol{\xi}_k \mid \mathcal{F}_{k-1}\right]}{\sqrt{\boldsymbol{A}_k}}\right\rangle = 0,$$

where we apply that $\mathbb{E}\left[\boldsymbol{\xi}_k \mid \mathcal{F}_{k-1}\right] = \mathbb{E}_{\boldsymbol{Z}_k}[\boldsymbol{\xi}_k] = 0$ from Assumption (A3). Then, using Assumption (A3) and Assumption (A4), we have

$$\|\bar{\boldsymbol{G}}_k\|_F = \|\mathbb{E}_{\boldsymbol{Z}_k}[\boldsymbol{G}_k]\|_F \leq \mathbb{E}_{\boldsymbol{Z}_k}\|\boldsymbol{G}_k\|_F \leq G, \quad \|\boldsymbol{\xi}_k\|_F = \|\boldsymbol{G}_k - \bar{\boldsymbol{G}}_k\|_F \leq 2G.$$

Let $\omega_k = 2G\eta_k \left\|\frac{\bar{\boldsymbol{G}}_k}{\sqrt{\boldsymbol{A}_k}}\right\|_F$. We thus derive from the Cauchy-Schwarz inequality that

$$\mathbb{E}\left[\exp\left(\frac{\zeta_k^2}{\omega_k^2}\right) \mid \mathcal{F}_{k-1}\right] \leq \mathbb{E}\left[\exp\left(\frac{\left\|\frac{\bar{\boldsymbol{G}}_k}{\sqrt{\boldsymbol{A}_k}}\right\|_F^2 \|\boldsymbol{\xi}_k\|_F^2}{4G^2 \left\|\frac{\bar{\boldsymbol{G}}_k}{\sqrt{\boldsymbol{A}_k}}\right\|_F^2}\right) \mid \mathcal{F}_{k-1}\right] \leq \exp(1).$$

Then, using Lemma B.1, it leads to that for any $\lambda > 0$, with probability at least $1 - \delta$,

$$-\sum_{k=1}^{t} \eta_k \left\langle \bar{\boldsymbol{G}}_k, \frac{\boldsymbol{\xi}_k}{\sqrt{\boldsymbol{A}_k}} \right\rangle \leq 3\lambda G^2 \sum_{k=1}^{t} \eta_k^2 \left\|\frac{\bar{\boldsymbol{G}}_k}{\sqrt{\boldsymbol{A}_k}}\right\|_F^2 + \frac{1}{\lambda}\log\left(\frac{1}{\delta}\right)$$

$$= 3\lambda G^2 \sum_{k=1}^{t}\sum_{i=1}^{n}\sum_{j=1}^{m} \frac{\eta_k}{\sqrt{a_{ij}^{(k)}}} \cdot \eta_k \frac{\left(\bar{g}_{ij}^{(k)}\right)^2}{\sqrt{a_{ij}^{(k)}}} + \frac{1}{\lambda}\log\left(\frac{1}{\delta}\right). \quad (57)$$

Meanwhile, when $\Theta_{\min} \leq \|\boldsymbol{X}_k\|_\infty \leq \Theta_{\max}$, $\rho_k = \rho_0/\sqrt{k}$, we have

$$\Theta_{\min} \leq \text{RMS}(\boldsymbol{X}_k) \leq \Theta_{\max}, \quad \frac{\max\{\epsilon_2, \Theta_{\min}\}\rho_0}{\sqrt{k}} \leq \eta_k \leq \frac{(\epsilon_2 + \Theta_{\max})\rho_0}{\sqrt{k}}. \quad (58)$$

Combining with Lemma B.3, we derive that

$$\frac{\eta_k}{\sqrt{a_{ij}^{(k)}}} \leq \frac{\eta_k}{\sqrt{\beta_{2,k}(1-\beta_{2,k})\epsilon_1}} \leq \frac{(\epsilon_2 + \Theta_{\max})\rho_0}{\sqrt{\beta_{2,k}\epsilon_1}} \cdot \frac{k^{c/2}}{\sqrt{k}} \quad (59)$$

$$\leq \frac{(\epsilon_2 + \Theta_{\max})\rho_0}{\sqrt{\min\{\beta_{2,1}, \beta_{2,2}\}\epsilon_1}} \leq \frac{2(\epsilon_2 + \Theta_{\max})\rho_0}{\sqrt{\epsilon_1}}, \quad (60)$$

where we use $\beta_{2,1} = 1/2, \beta_{2,2} = 1 - 1/2^c \geq 1 - 1/\sqrt{2}, c \in [1/2, 1]$ from (6) in the last inequality. Hence, plugging (60) into (57) and then re-scaling the $\delta$, we found that with probability at least $1 - \delta$, for all $t \in [T]$,

$$-\sum_{k=1}^{t} \eta_k \left\langle \bar{\boldsymbol{G}}_k, \frac{\boldsymbol{\xi}_k}{\sqrt{\boldsymbol{A}_k}} \right\rangle \leq \frac{6\lambda G^2(\epsilon_2 + \Theta_{\max})\rho_0}{\sqrt{\epsilon_1}} \sum_{k=1}^{t} \eta_k \left\|\frac{\bar{\boldsymbol{G}}_k}{\sqrt[4]{\boldsymbol{A}_k}}\right\|_F^2 + \frac{1}{\lambda}\log\left(\frac{T}{\delta}\right).$$

Setting $\lambda = \sqrt{\epsilon_1}/(24G^2(\epsilon_2 + \Theta_{\max})\rho_0)$, we derive the desired result. $\qquad\square$

The following key lemma provides an upper bound for the error brought by the proxy step-size $a_{ij}^{(k)}$, illustrating the error is controllable.

**Lemma B.7.** *For any $k \geq 1, i \in [n], j \in [m]$, it holds that*

$$\frac{\left|w_{ij}^{(k)} - a_{ij}^{(k)}\right|}{\sqrt{a_{ij}^{(k)}}} \leq \sqrt{1 - \beta_{2,k}}\min\{4\sqrt{\mathcal{G}}, G_1 + G_2\}, \quad (61)$$

*where $\mathcal{G}$ is as in (13) and $G_1, G_2$ are as in (44).*

*Proof.* To simplify the notation, we let

$$X = \beta_{2,k}R_{\boldsymbol{V}_{k-1}}^{(i)} + (1 - \beta_{2,k})R_{\boldsymbol{G}_{k,\epsilon_1}^2}^{(i)}, \quad \Delta X = (1 - \beta_{2,k})(\mathcal{G}_1 - R_{\boldsymbol{G}_{k,\epsilon_1}^2}^{(i)}),$$

$$Y = \beta_{2,k}C_{\boldsymbol{V}_{k-1}}^{(j)} + (1 - \beta_{2,k})C_{\boldsymbol{G}_{k,\epsilon_1}^2}^{(j)}, \quad \Delta Y = (1 - \beta_{2,k})(\mathcal{G}_2 - C_{\boldsymbol{G}_{k,\epsilon_1}^2}^{(j)}),$$

$$Z = \beta_{2,k}S_{\boldsymbol{V}_{k-1}} + (1 - \beta_{2,k})S_{\boldsymbol{G}_{k,\epsilon_1}^2}, \quad \Delta Z = (1 - \beta_{2,k})(\mathcal{G} - S_{\boldsymbol{G}_{k,\epsilon_1}^2}). \quad (62)$$

Then we have
$$\left|w_{ij}^{(k)} - a_{ij}^{(k)}\right| = \left|\frac{XY}{Z} - \frac{(X+\Delta X)(Y+\Delta Y)}{Z+\Delta Z}\right| = \left|\frac{XY\Delta Z - XZ\Delta Y - YZ\Delta X - Z(\Delta X\Delta Y)}{Z(Z+\Delta Z)}\right|.$$

Applying Lemma B.2, we could verify that $X, Y, Z \geq 0$ and
$$0 \leq \Delta X \leq (1-\beta_{2,k})\mathcal{G}_1, \quad 0 \leq \Delta Y \leq (1-\beta_{2,k})\mathcal{G}_2, \quad 0 \leq \Delta Z \leq (1-\beta_{2,k})\mathcal{G}. \quad (63)$$

Hence, we derive that
$$\frac{\left|w_{ij}^{(k)} - a_{ij}^{(k)}\right|}{\sqrt{a_{ij}^{(k)}}} = \frac{|XY\Delta Z - XZ\Delta Y - YZ\Delta X - Z(\Delta X\Delta Y)|}{Z\sqrt{(X+\Delta X)(Y+\Delta Y)(Z+\Delta Z)}}$$
$$\leq \underbrace{\frac{|X\Delta Y + Y\Delta X + (\Delta X\Delta Y)|}{\sqrt{(X+\Delta X)(Y+\Delta Y)(Z+\Delta Z)}}}_{\text{(I)}} + \underbrace{\frac{XY\Delta Z}{Z\sqrt{(X+\Delta X)(Y+\Delta Y)(Z+\Delta Z)}}}_{\text{(II)}}.$$
$$(64)$$

Since $XY \geq 0$ from (62), Term **(I)** could be bounded as
$$\textbf{(I)} \leq \frac{|X\Delta Y + Y\Delta X + (\Delta X\Delta Y)|}{\sqrt{(X\Delta Y + Y\Delta X + (\Delta X\Delta Y))(Z+\Delta Z)}} \leq \sqrt{\frac{X\Delta Y + Y\Delta X + (\Delta X\Delta Y)}{Z+\Delta Z}}. \quad (65)$$

Recalling the definition, we have $R_{\boldsymbol{V}_{k-1}}^{(i)} \leq S_{\boldsymbol{V}_{k-1}}, C_{\boldsymbol{V}_{k-1}}^{(j)} \leq S_{\boldsymbol{V}_{k-1}}$ for any $i \in [n], j \in [m]$. Further, applying Lemma B.2 and (63), we derive that
$$\frac{X\Delta Y}{Z+\Delta Z} \leq \left(\frac{R_{\boldsymbol{V}_{k-1}}^{(i)}}{S_{\boldsymbol{V}_{k-1}}} + \frac{R_{\boldsymbol{G}_{k,\epsilon_1}^2}^{(i)}}{\mathcal{G}}\right)\Delta Y \leq 2(1-\beta_{2,k})\mathcal{G}_2.$$

$$\frac{Y\Delta X}{Z+\Delta Z} \leq \left(\frac{C_{\boldsymbol{V}_{k-1}}^{(j)}}{S_{\boldsymbol{V}_{k-1}}} + \frac{C_{\boldsymbol{G}_{k,\epsilon_1}^2}^{(j)}}{\mathcal{G}}\right)\Delta X \leq 2(1-\beta_{2,k})\mathcal{G}_1,$$

$$\frac{\Delta X\Delta Y}{Z+\Delta Z} \leq \frac{\Delta X(1-\beta_{2,k})\mathcal{G}}{(1-\beta_{2,k})\mathcal{G}} \leq (1-\beta_{2,k})\mathcal{G}_1.$$

We then derive from (65), $\mathcal{G}_1 \leq \mathcal{G}$ and $\mathcal{G}_2 \leq \mathcal{G}$ that
$$\textbf{(I)} \leq \sqrt{5(1-\beta_{2,k})\mathcal{G}}. \quad (66)$$

To derive a free dimension bound, we could obtain from Lemma B.2, (63) and $\mathcal{G} \geq mn\epsilon_1/2$ that $Z + \Delta Z \geq mn\epsilon_1/2$. Hence,
$$\frac{X\Delta Y}{Z+\Delta Z} \leq \frac{2(1-\beta_{2,k})\mathcal{G}_1\mathcal{G}_2}{mn\epsilon_1}, \quad \frac{Y\Delta X}{Z+\Delta Z} \leq \frac{2(1-\beta_{2,k})\mathcal{G}_1\mathcal{G}_2}{mn\epsilon_1}, \quad \frac{\Delta X\Delta Y}{Z+\Delta Z} \leq \frac{2(1-\beta_{2,k})\mathcal{G}_1\mathcal{G}_2}{mn\epsilon_1}.$$

We then derive that
$$\textbf{(I)} \leq \sqrt{\frac{6(1-\beta_{2,k})\mathcal{G}_1\mathcal{G}_2}{mn\epsilon_1}} = \sqrt{\frac{6(1-\beta_{2,k})(G^4 + G^2\epsilon_1(m+n) + mn\epsilon_1^2)}{mn\epsilon_1}} \leq G_1\sqrt{1-\beta_{2,k}}, \quad (67)$$

where we used $m + n \leq mn$, and $G_1$ is defined in (44). Then, combining with (66) and (67), we have
$$\textbf{(I)} \leq \sqrt{1-\beta_{2,k}}\min\{\sqrt{5\mathcal{G}}, G_1\}, \quad (68)$$

where we applied that $m + n \leq mn$ when $m, n \geq 2$. Then we move to bound **(II)**. Recalling the definitions in (62), we have $X \leq Z, Y \leq Z$. Applying (63), we have
$$\textbf{(II)} \leq \frac{XY\Delta Z}{Z\sqrt{XY\Delta Z}} \leq \frac{\sqrt{XY\Delta Z}}{Z} \leq \sqrt{\Delta Z} \leq \sqrt{(1-\beta_{2,k})\mathcal{G}}.$$

Similarly, we derive from Lemma B.2 that $Z \geq mn\epsilon_1/2$, $X \leq \mathcal{G}_1$, $Y \leq \mathcal{G}_2$. Hence,

$$\textbf{(II)} \leq \frac{\sqrt{XY\Delta Z}}{Z} \leq \frac{2\sqrt{(1-\beta_{2,k})\mathcal{G}_1\mathcal{G}_2\mathcal{G}}}{mn\epsilon_1}$$

$$\leq 2\sqrt{1-\beta_{2,k}}\left(\frac{G^3}{mn\epsilon_1} + \frac{2G^2}{\sqrt{mn\epsilon_1}} + G + \frac{G}{\sqrt{mn}} + \sqrt{\epsilon_1}\right) \leq G_2\sqrt{1-\beta_{2,k}},$$

where $G_2$ has been defined in (44). We thus derive that

$$\textbf{(II)} \leq \sqrt{1-\beta_{2,k}}\min\{\sqrt{\mathcal{G}}, G_2\}. \tag{69}$$

Combining (68) with (69), we then derive the desired result. $\qquad\square$

### B.3 Proof of Proposition B.1

Using the inequality in (14), we have

$$f(\boldsymbol{X}_{k+1}) \leq f(\boldsymbol{X}_k) + \langle \bar{\boldsymbol{G}}_k, \boldsymbol{X}_{k+1} - \boldsymbol{X}_k\rangle + \frac{L}{2}\|\boldsymbol{X}_{k+1} - \boldsymbol{X}_k\|_F^2$$

$$\leq f(\boldsymbol{X}_k) - \eta_k\left\langle \bar{\boldsymbol{G}}_k, \frac{\boldsymbol{G}_k}{\sqrt{\boldsymbol{W}_k}}\right\rangle + \frac{L\eta_k^2}{2}\left\|\frac{\boldsymbol{G}_k}{\sqrt{\boldsymbol{W}_k}}\right\|_F^2.$$

Introducing the proxy step-size matrix $\boldsymbol{A}_k$ in (50) and then summing up both sides over $k \in [t]$, we derive that

$$f(\boldsymbol{X}_{t+1}) \leq f(\boldsymbol{X}_1) \underbrace{- \sum_{k=1}^{t}\eta_k\left\langle \bar{\boldsymbol{G}}_k, \frac{\boldsymbol{G}_k}{\sqrt{\boldsymbol{A}_k}}\right\rangle}_{\textbf{A}}$$

$$+ \underbrace{\sum_{k=1}^{t}\eta_k\left\langle \bar{\boldsymbol{G}}_k, \boldsymbol{G}_k \odot \left(\frac{1}{\sqrt{\boldsymbol{A}_k}} - \frac{1}{\sqrt{\boldsymbol{W}_k}}\right)\right\rangle}_{\textbf{B}} + \underbrace{\sum_{k=1}^{t}\frac{L\eta_k^2}{2}\left\|\frac{\boldsymbol{G}_k}{\sqrt{\boldsymbol{W}_k}}\right\|_F^2}_{\textbf{C}}. \tag{70}$$

**Estimation for A**   We first introduce $\boldsymbol{\xi}_k$ into **A**,

$$\textbf{A} = -\sum_{k=1}^{t}\eta_k\left\|\frac{\bar{\boldsymbol{G}}_k}{\sqrt[4]{\boldsymbol{A}_k}}\right\|_F^2 - \sum_{k=1}^{t}\eta_k\left\langle \bar{\boldsymbol{G}}_k, \frac{\boldsymbol{\xi}_k}{\sqrt{\boldsymbol{A}_k}}\right\rangle. \tag{71}$$

Then, using Lemma B.6, with probability at least $1 - \delta$, for all $t \in [T]$,

$$\textbf{A} = -\frac{3}{4}\sum_{k=1}^{t}\eta_k\left\|\frac{\bar{\boldsymbol{G}}_k}{\sqrt[4]{\boldsymbol{A}_k}}\right\|_F^2 + \frac{24G^2(\epsilon_2 + \Theta_{\max})\rho_0}{\sqrt{\epsilon_1}}\log\left(\frac{T}{\delta}\right). \tag{72}$$

**Estimation for B**   Term **B** is essentially the error brought by the proxy step-size $\boldsymbol{A}_k$. We will first calculate the gap of $1/\sqrt{w_{ij}^{(k)}}$ and $1/\sqrt{a_{ij}^{(k)}}$ as follows,

$$\left|\frac{1}{\sqrt{w_{ij}^{(k)}}} - \frac{1}{\sqrt{a_{ij}^{(k)}}}\right| = \frac{1}{\sqrt{w_{ij}^{(k)}}\sqrt{a_{ij}^{(k)}}}\left|\sqrt{w_{ij}^{(k)}} - \sqrt{a_{ij}^{(k)}}\right| \leq \frac{1}{\sqrt{w_{ij}^{(k)}}\sqrt{a_{ij}^{(k)}}}\sqrt{\left|w_{ij}^{(k)} - a_{ij}^{(k)}\right|}. \tag{73}$$

We then apply (73) and Young's inequality,

$$\textbf{B} \leq \sum_{k=1}^{t}\sum_{i=1}^{n}\sum_{j=1}^{m}\eta_k\left|\bar{g}_{ij}^{(k)}g_{ij}^{(k)}\right|\left|\frac{1}{\sqrt{w_{ij}^{(k)}}} - \frac{1}{\sqrt{a_{ij}^{(k)}}}\right|$$

$$\leq \sum_{k=1}^{t}\sum_{i=1}^{n}\sum_{j=1}^{m}\eta_k\frac{\left|\bar{g}_{ij}^{(k)}g_{ij}^{(k)}\right|}{\sqrt{w_{ij}^{(k)}}\sqrt{a_{ij}^{(k)}}}\sqrt{\left|w_{ij}^{(k)} - a_{ij}^{(k)}\right|}$$

$$\leq \frac{1}{4}\sum_{k=1}^{t}\sum_{i=1}^{n}\sum_{j=1}^{m}\eta_k \cdot \frac{\left(\bar{g}_{ij}^{(k)}\right)^2}{\sqrt{a_{ij}^{(k)}}} + 4\sum_{k=1}^{t}\sum_{i=1}^{n}\sum_{j=1}^{m}\eta_k \cdot \frac{\left|w_{ij}^{(k)} - a_{ij}^{(k)}\right|}{\sqrt{a_{ij}^{(k)}}}\cdot\left(\frac{g_{ij}^{(k)}}{\sqrt{w_{ij}^{(k)}}}\right)^2. \tag{74}$$

Thus, plugging (61) in Lemma B.7 into (74), we derive that

$$
\begin{aligned}
\mathbf{B} &\leq \frac{1}{4}\sum_{k=1}^{t}\eta_k\left\|\frac{\bar{\boldsymbol{G}}_k}{\sqrt[4]{\boldsymbol{A}_k}}\right\|_F^2 + 4\sqrt{\mathcal{G}}\sum_{k=1}^{t}\eta_k\sqrt{1-\beta_{2,k}}\left\|\frac{\boldsymbol{G}_k}{\sqrt{\boldsymbol{W}_k}}\right\|_F^2 \\
&\leq \frac{1}{4}\sum_{k=1}^{t}\eta_k\left\|\frac{\bar{\boldsymbol{G}}_k}{\sqrt[4]{\boldsymbol{A}_k}}\right\|_F^2 + 4\sqrt{\mathcal{G}}\sum_{k=1}^{t}\frac{(\epsilon_2+\Theta_{\max})\rho_0}{\sqrt{k}}\sqrt{1-\beta_{2,k}}\left\|\frac{\boldsymbol{G}_k}{\sqrt{\boldsymbol{W}_k}}\right\|_F^2 \\
&\leq \frac{1}{4}\sum_{k=1}^{t}\eta_k\left\|\frac{\bar{\boldsymbol{G}}_k}{\sqrt[4]{\boldsymbol{A}_k}}\right\|_F^2 + 4\sqrt{\mathcal{G}}\sum_{k=1}^{t}(\epsilon_2+\Theta_{\max})\rho_0(1-\beta_{2,k})\left\|\frac{\boldsymbol{G}_k}{\sqrt{\boldsymbol{W}_k}}\right\|_F^2,
\end{aligned}
\tag{75}
$$

where we used (58) in the second inequality and $1/\sqrt{k}\leq 1/k^{c/2}, c\in[1/2,1]$. Furthermore, using Lemma B.4 and Lemma B.5, we derive that

$$
\mathbf{B} \leq \frac{1}{4}\sum_{k=1}^{t}\eta_k\left\|\frac{\bar{\boldsymbol{G}}_k}{\sqrt[4]{\boldsymbol{A}_k}}\right\|_F^2 + \frac{8mn\mathcal{G}^{\frac{3}{2}}(\epsilon_2+\Theta_{\max})\rho_0}{\max\{m,n\}\epsilon_1}\left[\log\left(2+\frac{2G^2}{\epsilon_1}\right)+4\sum_{k=1}^{t}(1-\beta_{2,k})\right]. \tag{76}
$$

**Estimating C**   Using the similar deduction in (75) and (76), we derive that

$$
\mathbf{C} \leq \frac{Lmn\mathcal{G}(\epsilon_2+\Theta_{\max})^2\rho_0^2}{\max\{m,n\}\epsilon_1}\left[\log\left(2+\frac{2G^2}{\epsilon_1}\right)+4\sum_{k=1}^{t}(1-\beta_{2,k})\right]. \tag{77}
$$

**Putting together**   We first re-arrange the order in (70) and use $f(\boldsymbol{X}_{t+1})\geq f^*$ in Assumption (A2) to derive that

$$
0 \leq f(\boldsymbol{X}_1)-f^*+\mathbf{A}+\mathbf{B}+\mathbf{C}. \tag{78}
$$

We then plug (72), (76), (77) into (78) and set $t=T$, which leads to that with probability at least $1-\delta$,

$$
\frac{1}{2}\sum_{k=1}^{T}\eta_k\left\|\frac{\bar{\boldsymbol{G}}_k}{\sqrt[4]{\boldsymbol{A}_k}}\right\|_F^2 \leq C_1\log\left(\frac{T}{\delta}\right)+C_2\sum_{k=1}^{T}(1-\beta_{2,k})+C_3, \tag{79}
$$

where $C_1, C_2, C_3$ are as in Theorem B.1. Moreover, using Lemma B.3 and (58), we have

$$
\frac{1}{2}\sum_{k=1}^{T}\eta_k\left\|\frac{\bar{\boldsymbol{G}}_k}{\sqrt[4]{\boldsymbol{A}_k}}\right\|_F^2 \geq \sum_{k=1}^{T}\frac{\eta_k\left\|\bar{\boldsymbol{G}}_k\right\|_F^2}{2\max_{i,j}\sqrt{a_{ij}^{(k)}}} \geq \frac{\rho_0\max\{\epsilon_2,\Theta_{\min}\}}{2\sqrt{2\mathcal{G}}}\sum_{k=1}^{T}\frac{\left\|\bar{\boldsymbol{G}}_k\right\|_F^2}{\sqrt{k}}. \tag{80}
$$

Combining with (80) and (79), and using $\sum_{k=1}^{T}1/\sqrt{k}\geq\sqrt{T}$, we derive that

$$
\min_{k\in[T]}\|\bar{\boldsymbol{G}}_k\|^2 \leq \frac{C_0}{\sqrt{T}}\left(C_1\log\left(\frac{T}{\delta}\right)+C_2\sum_{k=1}^{T}(1-\beta_{2,k})+C_3\right), \tag{81}
$$

where $C_0$ has already been defined in (42). We then derive the first desired result that

$$
\min_{k\in[T]}\|\bar{\boldsymbol{G}}_k\|^2 \leq \frac{C_0}{\sqrt{T}}\left(C_1\log\left(\frac{T}{\delta}\right)+C_2\sum_{k=1}^{T}\frac{1}{k^c}+C_3\right).
$$

**Free dimension bound**   We follow the similar deduction in (75) and use Lemma B.7 to derive that

$$
\mathbf{B} \leq \frac{1}{4}\sum_{k=1}^{t}\eta_k\left\|\frac{\bar{\boldsymbol{G}}_k}{\sqrt[4]{\boldsymbol{A}_k}}\right\|_F^2 + 4(G_1+G_2)(\epsilon_2+\Theta_{\max})\rho_0\sum_{k=1}^{t}\frac{1}{k^{c/2+1/2}}\left\|\frac{\boldsymbol{G}_k}{\sqrt{\boldsymbol{W}_k}}\right\|_F^2. \tag{82}
$$

Recalling the definition of $w_{ij}^{(k)}$ in (49) and Lemma B.2, we derive that

$$
w_{ij}^{(k)} = \frac{R_{\boldsymbol{V}_k}^{(i)}C_{\boldsymbol{V}_k}^{(j)}}{S_{\boldsymbol{V}_k}} \geq \frac{mn\epsilon_1^2}{4\mathcal{G}}, \quad \left\|\frac{\boldsymbol{G}_k}{\sqrt{\boldsymbol{W}_k}}\right\|_F^2 \leq \frac{\|\boldsymbol{G}_k\|_F^2}{\min_{i,j}w_{ij}^{(k)}} \leq \frac{4G^2\mathcal{G}}{mn\epsilon_1^2} \leq G_3, \tag{83}
$$

where $G_3$ is as in (44). We thus derive from (82) and (83) that

$$\mathbf{B} \le \frac{1}{4} \sum_{k=1}^{t} \eta_k \left\| \frac{\bar{\boldsymbol{G}}_k}{\sqrt[4]{\boldsymbol{A}_k}} \right\|_F^2 + 4G_3(G_1 + G_2)(\epsilon_2 + \Theta_{\max})\rho_0 \sum_{k=1}^{t} \frac{1}{k^{c/2+1/2}}. \tag{84}$$

Using (58) and (83), we derive that

$$\mathbf{C} = \sum_{k=1}^{t} \frac{L\eta_k^2}{2} \left\| \frac{\boldsymbol{G}_k}{\sqrt{\boldsymbol{W}_k}} \right\|_F^2 \le \frac{LG_3(\epsilon_2 + \Theta_{\max})^2\rho_0^2}{2} \sum_{k=1}^{t} \frac{1}{k}. \tag{85}$$

Plugging the unchanged estimation for $\mathbf{A}$ in (72), (84) and (85) into (70), we have that with probability at least $1 - \delta$, for all $t \in [T]$,

$$\frac{1}{2} \sum_{k=1}^{t} \eta_k \left\| \frac{\bar{\boldsymbol{G}}_k}{\sqrt[4]{\boldsymbol{A}_k}} \right\|_F^2 \le C_1 \log\left(\frac{T}{\delta}\right) + C_2' \sum_{k=1}^{t} \frac{1}{k^{c/2+1/2}} + C_3' \sum_{k=1}^{t} \frac{1}{k}, \tag{86}$$

where $C_2', C_3'$ are given as in (43) and $C_1$ is as in (41). Further, using Lemma B.3 and the similar deduction for (80),

$$\frac{1}{2} \sum_{k=1}^{t} \eta_k \left\| \frac{\bar{\boldsymbol{G}}_k}{\sqrt[4]{\boldsymbol{A}_k}} \right\|_F^2 \ge \sum_{k=1}^{t} \frac{\eta_k \left\| \bar{\boldsymbol{G}}_k \right\|_F^2}{2 \max_{i,j} \sqrt{a_{ij}^{(k)}}} \ge \frac{1}{C_0'} \sum_{k=1}^{t} \frac{\left\| \bar{\boldsymbol{G}}_k \right\|_F^2}{\sqrt{k}}, \tag{87}$$

where $C_0'$ is as in (43). Combining with (86) and (87), and setting $t = T$, we derive the second desired result in Proposition B.1 that

$$\min_{k \in [T]} \|\bar{\boldsymbol{G}}_k\|^2 \le \frac{C_0'}{\sqrt{T}} \left( C_1 \log\left(\frac{T}{\delta}\right) + C_2' \sum_{k=1}^{T} \frac{1}{k^{c/2+1/2}} + C_3' \sum_{k=1}^{T} \frac{1}{k} \right).$$

### B.4  Proof of Theorem B.1

Now based on the result in Proposition B.1, we could further derive the final convergence rate. Noting that when $c = 1$, we could bound that

$$\sum_{k=1}^{T} \frac{1}{k} \le 1 + \int_1^T \frac{1}{x} dx \le 1 + \log T. \tag{88}$$

Then, we obtain that

$$\min_{k \in [T]} \|\bar{\boldsymbol{G}}_k\|_F^2 \le \frac{C_0}{\sqrt{T}} \left( C_1 \log\left(\frac{T}{\delta}\right) + C_2 \log T + C_2 + C_3 \right),$$

$$\min_{k \in [T]} \|\bar{\boldsymbol{G}}_k\|_F^2 \le \frac{C_0'}{\sqrt{T}} \left( C_1 \log\left(\frac{T}{\delta}\right) + (C_2' + C_3') \log T + C_2' + C_3' \right).$$

When $1/2 \le c < 1$, we have

$$\sum_{k=1}^{T} \frac{1}{k^c} \le 1 + \int_1^T \frac{1}{x^c} dx \le 1 + \frac{T^{1-c}}{1-c},$$

$$\sum_{k=1}^{T} \frac{1}{k^{c/2+1/2}} \le 1 + \int_1^T \frac{1}{x^{c/2+1/2}} dx \le 1 + \frac{2T^{(1-c)/2}}{1-c}. \tag{89}$$

Then, we obtain that

$$\min_{k \in [T]} \|\bar{\boldsymbol{G}}_k\|_F^2 \le \frac{C_0}{\sqrt{T}} \left( C_1 \log\left(\frac{T}{\delta}\right) + \frac{C_2}{1-c} \cdot T^{1-c} + C_2 + C_3 \right),$$

$$\min_{k \in [T]} \|\bar{\boldsymbol{G}}_k\|_F^2 \le \frac{C_0}{\sqrt{T}} \left( C_1 \log\left(\frac{T}{\delta}\right) + \frac{2C_2'}{1-c} \cdot T^{1-c} + C_3' \log T + C_2' + C_3' \right).$$

## C  Proof detail for stochastic Adafactor with update clipping

We first provide the detailed version of Theorem 7.1 as follows.

**Theorem C.1.** *Let $\{\boldsymbol{X}_k\}_{k\geq 1}$ be the sequence generated by Algorithm 1 with (7). If Assumptions (A1)-(A4) hold, and*

$$\rho_k = \rho_0/\sqrt{k}, \quad d_k = k^{\frac{c}{2(\alpha-1)}}, \quad \forall k \geq 1,$$
$$\beta_{2,1} = 1/2, \quad \beta_{2,k} = 1 - 1/k^c, \forall k \geq 2.$$

*When $c = 1$, with probability at least $1 - \delta$,*

$$\min_{k\in[T]} \|\bar{\boldsymbol{G}}_k\|_F^2 \leq \frac{D_0}{\sqrt{T}} \left( C_1 \log\left(\frac{T}{\delta}\right) + (C_2 + D_1(\alpha)) \log T + C_2 + D_1(\alpha) + C_3 \right), \tag{90}$$

$$\min_{k\in[T]} \|\bar{\boldsymbol{G}}_k\|_F^2 \leq \frac{D_0}{\sqrt{T}} \left( C_1 \log\left(\frac{T}{\delta}\right) + (C_2' + C_3' + D_1(\alpha)) \log T + C_2' + C_3' + D_1(\alpha) \right). \tag{91}$$

*When $1/2 \leq c < 1$, with probability at least $1 - \delta$,*

$$\min_{k\in[T]} \|\bar{\boldsymbol{G}}_k\|_F^2 \leq \frac{D_0}{\sqrt{T}} \left( C_1 \log\left(\frac{T}{\delta}\right) + \frac{C_2 + D_1(\alpha)}{1-c} \cdot T^{1-c} + C_2 + D_1(\alpha) + C_3 \right), \tag{92}$$

$$\min_{k\in[T]} \|\bar{\boldsymbol{G}}_k\|_F^2 \leq \frac{D_0}{\sqrt{T}} \left( C_1 \log\left(\frac{T}{\delta}\right) + C_3' \log T + \frac{2(C_2' + D_1(\alpha))}{1-c} \cdot T^{\frac{1-c}{2}} + C_2' + C_3' + D_1(\alpha) \right), \tag{93}$$

*where $C_1, C_2, C_3, C_2', C_3'$ are as in Theorem B.1 and*

$$D_0 = \min\{C_0, C_0'\}, \quad D_1(\alpha) = \frac{G^{1+\alpha}G_4^{1-\alpha}\sqrt{\mathcal{G}}(\epsilon_2 + \Theta_{\max})\rho_0}{\sqrt{mn}\epsilon_1}, \quad G_4 = \frac{mn\epsilon_1}{2\sqrt{\mathcal{G}}}. \tag{94}$$

**Calculation of hyper-parameters' dependency**  We first calculate the dependency on $m, n, \epsilon_1, \alpha$ in the additional coefficient $D_1(\alpha)$ as follows,

$$D_1(\alpha) \sim \mathcal{O}\left( \left(\frac{\sqrt{1 + mn\epsilon_1}}{mn\epsilon_1}\right)^{\alpha-1} \sqrt{\frac{1}{mn\epsilon_1^2} + \frac{1}{\epsilon_1}} \right), \tag{95}$$

which is free of the curse of dimension since $mn$ exists in the denominator. Recalling the definitions of $C_0', C_1, C_2', C_3'$ in (41) and (43), it's easy to verify that these coefficients are also free of the curse of dimension factor $m, n$ since $m, n$ exist in the denominator. Thereby, we also derive a free dimension bound selecting (91) and (93).

To calculate the dependency on $\epsilon_1$, we could combine with (45) and (95) to derive that

$$C_0 D_1(\alpha) \sim \mathcal{O}\left(\epsilon_1^{-\alpha}\right), \quad C_0 C_1 \sim \mathcal{O}\left(1/\epsilon_1^{-1/2}\right), \quad C_0 C_3 \sim \mathcal{O}\left(\epsilon_1^{-1} \log(1/\epsilon_1)\right).$$

Thereby, selecting the bounds in (90) and (92) and noting that $\alpha > 1$, we derive that the order on $\epsilon_1$ is

$$\mathcal{O}\left(\frac{1}{\epsilon_1^{\alpha}} \log\left(\frac{1}{\epsilon_1}\right)\right). \tag{96}$$

Moreover, it's clear to reveal that there exist $mn$ in denominator, which could improve the dependency on $\epsilon_1$. If we suppose that $mn$ is comparable to $\epsilon_1$, then we derive that $C_0 D_1(\alpha) \sim \mathcal{O}(\epsilon_1^{-1/2})$ and the order on $\epsilon_1$ is

$$\mathcal{O}\left(\frac{1}{\epsilon_1} \log\left(\frac{1}{\epsilon_1}\right)\right). \tag{97}$$

### C.1  Proof of Theorem C.1

We define

$$\tilde{\boldsymbol{G}}_k = \frac{\boldsymbol{G}_k}{\max\{1, \|\boldsymbol{U}_k\|_F/(d_k\sqrt{mn})\}}, \quad \hat{\rho}_k = \max\{\epsilon_2, \mathrm{RMS}(\boldsymbol{X}_k)\}\rho_k. \tag{98}$$

674 Since $\text{RMS}(\boldsymbol{U}_k) = \|\boldsymbol{U}_k\|_F/\sqrt{mn}$, $\Theta_{\min} \leq \text{RMS}(\boldsymbol{X}_k) \leq \Theta_{\max}$, we derive that

$$\boldsymbol{X}_{k+1} = \boldsymbol{X}_k - \hat{\rho}_k \frac{\tilde{\boldsymbol{G}}_k}{\sqrt{\boldsymbol{W}_k}},$$

$$\frac{\max\{\epsilon_2, \Theta_{\min}\}\rho_0}{\sqrt{k}} \leq \hat{\rho}_k \leq \frac{(\epsilon_2 + \Theta_{\max})\rho_0}{\sqrt{k}} \leq (\epsilon_2 + \Theta_{\max})\rho_0\sqrt{1 - \beta_{2,k}}, \tag{99}$$

675 where we applied that $1/\sqrt{k} \leq 1/k^{c/2}, c \in [1/2, 1]$ and $\beta_{2,k} = 1 - 1/k^c$ in the last inequality. Using
676 the inequalities in (14) and (99), we have

$$f(\boldsymbol{X}_{k+1}) \leq f(\boldsymbol{X}_k) + \langle \bar{\boldsymbol{G}}_k, \boldsymbol{X}_{k+1} - \boldsymbol{X}_k \rangle + \frac{L}{2}\|\boldsymbol{X}_{k+1} - \boldsymbol{X}_k\|_F^2$$

$$\leq f(\boldsymbol{X}_k) - \hat{\rho}_k \left\langle \bar{\boldsymbol{G}}_k, \frac{\tilde{\boldsymbol{G}}_k}{\sqrt{\boldsymbol{W}_k}} \right\rangle + \frac{L\hat{\rho}_k^2}{2} \left\| \frac{\tilde{\boldsymbol{G}}_k}{\sqrt{\boldsymbol{W}_k}} \right\|_F^2.$$

677 Summing up both sides over $k \in [t]$ and using $f(\boldsymbol{X}_{t+1}) \geq f^*$ from Assumption (A2), we derive that

$$0 \leq f(\boldsymbol{X}_1) - f^* + \underbrace{\sum_{k=1}^t -\hat{\rho}_k \left\langle \bar{\boldsymbol{G}}_k, \frac{\tilde{\boldsymbol{G}}_k}{\sqrt{\boldsymbol{W}_k}} \right\rangle}_{\mathbf{D}} + \underbrace{\sum_{k=1}^t \frac{L\hat{\rho}_k^2}{2} \left\| \frac{\tilde{\boldsymbol{G}}_k}{\sqrt{\boldsymbol{W}_k}} \right\|_F^2}_{\mathbf{E}}. \tag{100}$$

678 Introducing $\boldsymbol{A}_k$ in (50), we further have the following decomposition,

$$\mathbf{D} = -\sum_{k=1}^t \hat{\rho}_k \left\langle \bar{\boldsymbol{G}}_k, \frac{\tilde{\boldsymbol{G}}_k}{\sqrt{\boldsymbol{A}_k}} \right\rangle + \underbrace{\sum_{k=1}^t \hat{\rho}_k \left\langle \bar{\boldsymbol{G}}_k, \left( \frac{1}{\sqrt{\boldsymbol{A}_k}} - \frac{1}{\sqrt{\boldsymbol{W}_k}} \right) \odot \tilde{\boldsymbol{G}}_k \right\rangle}_{\mathbf{D.1}}$$

$$= -\sum_{k=1}^t \hat{\rho}_k \left\| \frac{\bar{\boldsymbol{G}}_k}{\sqrt[4]{\boldsymbol{A}_k}} \right\|_F^2 + \mathbf{D.1}$$

$$\underbrace{-\sum_{k=1}^t \hat{\rho}_k \left\langle \bar{\boldsymbol{G}}_k, \frac{\tilde{\boldsymbol{G}}_k}{\sqrt{\boldsymbol{A}_k}} - \mathbb{E}_{\boldsymbol{Z}_k}\left[ \frac{\tilde{\boldsymbol{G}}_k}{\sqrt{\boldsymbol{A}_k}} \right] \right\rangle}_{\mathbf{D.2}} + \underbrace{\sum_{k=1}^t \hat{\rho}_k \left\langle \bar{\boldsymbol{G}}_k, \frac{\bar{\boldsymbol{G}}_k}{\sqrt{\boldsymbol{A}_k}} - \mathbb{E}_{\boldsymbol{Z}_k}\left[ \frac{\tilde{\boldsymbol{G}}_k}{\sqrt{\boldsymbol{A}_k}} \right] \right\rangle}_{\mathbf{D.3}}. \tag{101}$$

679 **Estimating E**   Hence, using (98), (99), Lemma B.4 and Lemma B.5, we derive that

$$\mathbf{E} \leq \frac{L}{2}\sum_{k=1}^t \hat{\rho}_k^2 \left\| \frac{\boldsymbol{G}_k}{\sqrt{\boldsymbol{W}_k}} \right\|_F^2 \leq \frac{L(\epsilon_2 + \Theta_{\max})^2\rho_0^2}{2}\sum_{k=1}^t (1 - \beta_{2,k}) \left\| \frac{\boldsymbol{G}_k}{\sqrt{\boldsymbol{W}_k}} \right\|_F^2$$

$$\leq \frac{Lmn\mathcal{G}(\epsilon_2 + \Theta_{\max})^2\rho_0^2}{\max\{m,n\}\epsilon_1} \left[ \log\left( 2 + \frac{2G^2}{\epsilon_1} \right) + 4\sum_{k=1}^t (1 - \beta_{2,k}) \right]. \tag{102}$$

680 To avoid the curse of dimension, we drive from (98) and (83) that

$$\left\| \frac{\tilde{\boldsymbol{G}}_k}{\sqrt{\boldsymbol{W}_k}} \right\|_F^2 = \frac{1}{(\max\{1, \|\boldsymbol{U}_k\|_F/(d_k\sqrt{mn})\})^2} \left\| \frac{\boldsymbol{G}_k}{\sqrt{\boldsymbol{W}_k}} \right\|_F^2 \leq \left\| \frac{\boldsymbol{G}_k}{\sqrt{\boldsymbol{W}_k}} \right\|_F^2 \leq G_3. \tag{103}$$

681 Then, using (99) and (103), we derive that

$$\mathbf{E} \leq \frac{LG_3(\epsilon_2 + \Theta_{\max})^2\rho_0^2}{2}\sum_{k=1}^t \frac{1}{k}. \tag{104}$$

**Estimating D.1** We could follow the similar deduction in (73) and (74) to derive that

$$
\begin{aligned}
\mathbf{D.1} &\leq \sum_{k=1}^{t}\sum_{i=1}^{n}\sum_{j=1}^{m}\hat{\rho}_k|\bar{g}_{ij}^{(k)}\tilde{g}_{ij}^{(k)}|\left|\frac{1}{\sqrt{w_{ij}^{(k)}}}-\frac{1}{\sqrt{a_{ij}^{(k)}}}\right| \\
&\leq \sum_{k=1}^{t}\sum_{i=1}^{n}\sum_{j=1}^{m}\hat{\rho}_k\frac{|\bar{g}_{ij}^{(k)}\tilde{g}_{ij}^{(k)}|}{\sqrt{w_{ij}^{(k)}}\sqrt{a_{ij}^{(k)}}}\sqrt{\left|w_{ij}^{(k)}-a_{ij}^{(k)}\right|} \\
&\leq \frac{1}{4}\sum_{k=1}^{t}\sum_{i=1}^{n}\sum_{j=1}^{m}\hat{\rho}_k\cdot\frac{\left(\bar{g}_{ij}^{(k)}\right)^2}{\sqrt{a_{ij}^{(k)}}}+4\sum_{k=1}^{t}\sum_{i=1}^{n}\sum_{j=1}^{m}\hat{\rho}_k\cdot\frac{\left|w_{ij}^{(k)}-a_{ij}^{(k)}\right|}{\sqrt{a_{ij}^{(k)}}}\cdot\left(\frac{\tilde{g}_{ij}^{(k)}}{\sqrt{w_{ij}^{(k)}}}\right)^2.
\end{aligned} \tag{105}
$$

Using Lemma B.7 and (105), we further derive that

$$
\begin{aligned}
\mathbf{D.1} &\leq \frac{1}{4}\sum_{k=1}^{t}\hat{\rho}_k\left\|\frac{\bar{\boldsymbol{G}}_k}{\sqrt[4]{\boldsymbol{A}_k}}\right\|_F^2+4\sqrt{\mathcal{G}}\sum_{k=1}^{t}\hat{\rho}_k\sqrt{1-\beta_{2,k}}\left\|\frac{\tilde{\boldsymbol{G}}_k}{\sqrt{\boldsymbol{W}_k}}\right\|_F^2 \\
&\leq \frac{1}{4}\sum_{k=1}^{t}\hat{\rho}_k\left\|\frac{\bar{\boldsymbol{G}}_k}{\sqrt[4]{\boldsymbol{A}_k}}\right\|_F^2+4\sqrt{\mathcal{G}}\sum_{k=1}^{t}\hat{\rho}_k\sqrt{1-\beta_{2,k}}\left\|\frac{\boldsymbol{G}_k}{\sqrt{\boldsymbol{W}_k}}\right\|_F^2.
\end{aligned}
$$

Using (99), Lemma B.4 and Lemma B.5, we further have

$$
\begin{aligned}
\mathbf{D.1} &\leq \frac{1}{4}\sum_{k=1}^{t}\hat{\rho}_k\left\|\frac{\bar{\boldsymbol{G}}_k}{\sqrt[4]{\boldsymbol{A}_k}}\right\|_F^2+4\sqrt{\mathcal{G}}(\epsilon_2+\Theta_{\max})\rho_0\sum_{k=1}^{t}(1-\beta_{2,k})\left\|\frac{\boldsymbol{G}_k}{\sqrt{\boldsymbol{W}_k}}\right\|_F^2 \\
&\leq \frac{1}{4}\sum_{k=1}^{t}\hat{\rho}_k\left\|\frac{\bar{\boldsymbol{G}}_k}{\sqrt[4]{\boldsymbol{A}_k}}\right\|_F^2+\frac{8mn\mathcal{G}^{\frac{3}{2}}(\epsilon_2+\Theta_{\max})\rho_0}{\max\{m,n\}\epsilon_1}\left[\log\left(2+\frac{2G^2}{\epsilon_1}\right)+4\sum_{k=1}^{t}(1-\beta_{2,k})\right].
\end{aligned} \tag{106}
$$

To avoid the curse of dimension, we apply Lemma B.7, (99) and (83) to derive that

$$
\begin{aligned}
\mathbf{D.1} &\leq \frac{1}{4}\sum_{k=1}^{t}\hat{\rho}_k\left\|\frac{\bar{\boldsymbol{G}}_k}{\sqrt[4]{\boldsymbol{A}_k}}\right\|_F^2+4(G_1+G_2)\sum_{k=1}^{t}\hat{\rho}_k\sqrt{1-\beta_{2,k}}\left\|\frac{\boldsymbol{G}_k}{\sqrt{\boldsymbol{W}_k}}\right\|_F^2 \\
&\leq \frac{1}{4}\sum_{k=1}^{t}\hat{\rho}_k\left\|\frac{\bar{\boldsymbol{G}}_k}{\sqrt[4]{\boldsymbol{A}_k}}\right\|_F^2+4(G_1+G_2)(\epsilon_2+\Theta_{\max})\rho_0\sum_{k=1}^{t}\frac{1}{k^{c/2+1/2}}\left\|\frac{\boldsymbol{G}_k}{\sqrt{\boldsymbol{W}_k}}\right\|_F^2 \\
&\leq \frac{1}{4}\sum_{k=1}^{t}\hat{\rho}_k\left\|\frac{\bar{\boldsymbol{G}}_k}{\sqrt[4]{\boldsymbol{A}_k}}\right\|_F^2+4G_3(G_1+G_2)(\epsilon_2+\Theta_{\max})\rho_0\sum_{k=1}^{t}\frac{1}{k^{c/2+1/2}}.
\end{aligned} \tag{107}
$$

**Estimating D.2** Since $\boldsymbol{A}_k$ is independent from $\boldsymbol{Z}_k$, it further leads to

$$
\mathbf{D.2} = -\sum_{k=1}^{t}\hat{\rho}_k\left\langle\frac{\bar{\boldsymbol{G}}_k}{\sqrt{\boldsymbol{A}_k}},\tilde{\boldsymbol{G}}_k-\mathbb{E}_{\boldsymbol{Z}_k}\left[\tilde{\boldsymbol{G}}_k\right]\right\rangle.
$$

Then, the deduction for estimating **D.2** follows the similar idea as in Lemma B.6, relying on a martingale difference sequence.

Let us set $\varphi_k = -\hat{\rho}_k\left\langle\frac{\bar{\boldsymbol{G}}_k}{\sqrt{\boldsymbol{A}_k}},\tilde{\boldsymbol{G}}_k-\mathbb{E}_{\boldsymbol{Z}_k}\left[\tilde{\boldsymbol{G}}_k\right]\right\rangle$ and the filtration $\mathcal{F}_k = \sigma\left(\boldsymbol{Z}_1,\cdots,\boldsymbol{Z}_k\right)$. Noting that $\hat{\rho}_k$, $\bar{\boldsymbol{G}}_k$ and $\boldsymbol{A}_k$ are dependent by $\mathcal{F}_{k-1}$. Since $\boldsymbol{\xi}_k$ is dependent by $\mathcal{F}_k$, we could prove that $\{\varphi_k\}_{k\geq 1}$ is a martingale difference sequence by showing that

$$
\mathbb{E}\left[\varphi_k\mid\mathcal{F}_{k-1}\right] = -\hat{\rho}_k\left\langle\frac{\bar{\boldsymbol{G}}_k}{\sqrt{\boldsymbol{A}_k}},\mathbb{E}_{\boldsymbol{Z}_k}\left[\tilde{\boldsymbol{G}}_k-\mathbb{E}_{\boldsymbol{Z}_k}[\tilde{\boldsymbol{G}}_k]\right]\right\rangle = 0.
$$

In addition, using Assumptions (A3), (A4) and Jensen's inequality, we have

$$
\|\tilde{\boldsymbol{G}}_k\|_F = \frac{\|\boldsymbol{G}_k\|_F}{\max\{1,\|\boldsymbol{U}_k\|/(d_k\sqrt{mn})\}} \leq \|\boldsymbol{G}_k\|_F \leq G,\quad \|\mathbb{E}_{\boldsymbol{Z}_k}[\tilde{\boldsymbol{G}}_k]\|_F \leq \mathbb{E}_{\boldsymbol{Z}_k}\|\tilde{\boldsymbol{G}}_k\|_F \leq G.
$$

Therefore, we derive that

$$\|\tilde{\boldsymbol{G}}_k - \mathbb{E}_{\boldsymbol{Z}_k}[\tilde{\boldsymbol{G}}_k]\|_F \le \|\tilde{\boldsymbol{G}}_k\|_F + \|\mathbb{E}_{\boldsymbol{Z}_k}[\tilde{\boldsymbol{G}}_k]\|_F \le 2G. \tag{108}$$

Let $\omega'_k = 2G\hat{\rho}_k \left\| \frac{\bar{\boldsymbol{G}}_k}{\sqrt{\boldsymbol{A}_k}} \right\|_F$. We thus derive from the Cauchy-Schwarz inequality and (108) that

$$\mathbb{E}\left[\exp\left(\frac{\varphi_k^2}{(\omega'_k)^2}\right) \mid \mathcal{F}_{k-1}\right] \le \mathbb{E}\left[\exp\left(\frac{\left\|\frac{\bar{\boldsymbol{G}}_k}{\sqrt{\boldsymbol{A}_k}}\right\|_F^2 \|\tilde{\boldsymbol{G}}_k - \mathbb{E}_{\boldsymbol{Z}_k}[\tilde{\boldsymbol{G}}_k]\|_F^2}{4G^2 \left\|\frac{\bar{\boldsymbol{G}}_k}{\sqrt{\boldsymbol{A}_k}}\right\|_F^2}\right) \mid \mathcal{F}_{k-1}\right] \le \exp(1).$$

Then, using Lemma B.1, it leads to that for any $\lambda > 0$, with probability at least $1 - \delta$,

$$\mathbf{D.2} = \sum_{k=1}^{t} \varphi_k \le 3\lambda G^2 \sum_{k=1}^{t} \hat{\rho}_k^2 \left\|\frac{\bar{\boldsymbol{G}}_k}{\sqrt{\boldsymbol{A}_k}}\right\|_F^2 + \frac{1}{\lambda}\log\left(\frac{1}{\delta}\right)$$

$$= 3\lambda G^2 \sum_{k=1}^{t}\sum_{i=1}^{n}\sum_{j=1}^{m} \frac{\hat{\rho}_k}{\sqrt{a_{ij}^{(k)}}} \cdot \hat{\rho}_k \frac{\left(\bar{g}_{ij}^{(k)}\right)^2}{\sqrt{a_{ij}^{(k)}}} + \frac{1}{\lambda}\log\left(\frac{1}{\delta}\right).$$

Since $\{\beta_{2,k}\}_{k\ge 2}$ is non-decreasing, we could apply Lemma B.3 to derive that

$$\frac{1}{\sqrt{a_{ij}^{(k)}}} \le \sqrt{\frac{1}{\beta_{2,k}(1 - \beta_{2,k})\epsilon_1}} \le \sqrt{\frac{1}{\min\{\beta_{2,1}, \beta_{2,2}\}(1 - \beta_{2,k})\epsilon_1}} \le \frac{2}{\sqrt{(1 - \beta_{2,k})\epsilon_1}}.$$

Then, we apply (99), and re-scale $\delta$ to obtain that for any $\lambda > 0$, with probability at least $1 - \delta$, for all $t \in [T]$,

$$\mathbf{D.2} \le \frac{6\lambda G^2 \rho_0(\epsilon_2 + \Theta_{\max})}{\sqrt{\epsilon_1}} \sum_{k=1}^{t} \hat{\rho}_k \left\|\frac{\bar{\boldsymbol{G}}_k}{\sqrt[4]{\boldsymbol{A}_k}}\right\|_F^2 + \frac{1}{\lambda}\log\left(\frac{T}{\delta}\right).$$

Setting $\lambda = \sqrt{\epsilon_1}/(24G^2\rho_0(\epsilon_2 + \Theta_{\max}))$, we derive that

$$\mathbf{D.2} \le \frac{1}{4}\sum_{k=1}^{t} \hat{\rho}_k \left\|\frac{\bar{\boldsymbol{G}}_k}{\sqrt[4]{\boldsymbol{A}_k}}\right\|_F^2 + \frac{24G^2\rho_0(\epsilon_2 + \Theta_{\max})}{\sqrt{\epsilon_1}}\log\left(\frac{T}{\delta}\right). \tag{109}$$

**Estimating D.3**  First, since $\boldsymbol{A}_k$ is independent from $\boldsymbol{Z}_k$ and $\mathbb{E}_{\boldsymbol{Z}_k}[\boldsymbol{G}_k] = \bar{\boldsymbol{G}}_k$, we have

$$\mathbf{D.3} = \sum_{k=1}^{t} \hat{\rho}_k \left\langle \bar{\boldsymbol{G}}_k, \frac{\mathbb{E}_{\boldsymbol{Z}_k}[\boldsymbol{G}_k]}{\sqrt{\boldsymbol{A}_k}} - \frac{\mathbb{E}_{\boldsymbol{Z}_k}[\tilde{\boldsymbol{G}}_k]}{\sqrt{\boldsymbol{A}_k}} \right\rangle$$

$$\le \sum_{k=1}^{t} \hat{\rho}_k \left\|\frac{\bar{\boldsymbol{G}}_k}{\sqrt{\boldsymbol{A}_k}}\right\|_F \cdot \left\|\mathbb{E}_{\boldsymbol{Z}_k}\left[\underbrace{\boldsymbol{G}_k - \frac{\boldsymbol{G}_k}{\max\{1, \|\boldsymbol{U}_k\|_F/(d_k\sqrt{mn})\}}}_{\Omega_k}\right]\right\|_F. \tag{110}$$

We define the random variable $S_k^{(1)}$, $S_k^{(2)}$ and $\tilde{S}_k^{(1)}$ using the indicator function $\chi$ and $G_4$ in (94) as follows,

$$S_k^{(1)} = \chi_{\{\|\boldsymbol{U}_k\|_F > d_k\sqrt{mn}\}}, \quad S_k^{(2)} = \chi_{\{\|\boldsymbol{U}_k\|_F \le d_k\sqrt{mn}\}}, \quad \tilde{S}_k^{(1)} = \chi_{\{\|\boldsymbol{G}_k\|_F \ge d_k G_4\}}.$$

From (83), we derive that

$$\|\boldsymbol{U}_k\|_F \le \|\boldsymbol{G}_k\|_F \cdot \frac{2\sqrt{\mathcal{G}}}{\sqrt{mn\epsilon_1}}.$$

Hence, $S_k^{(1)} \le \tilde{S}_k^{(1)}, \forall k \ge 1$. Note that when $S_k^{(2)} = 1$, it's equivalent to $\Omega_k = 0$. Then, we derive that

$$\|\mathbb{E}_{\boldsymbol{Z}_k}[\Omega_k]\|_F = \left\|\mathbb{E}_{\boldsymbol{Z}_k}[\Omega_k S_k^{(1)}] + \mathbb{E}_{\boldsymbol{Z}_k}[\Omega_k S_k^{(2)}]\right\|_F = \left\|\mathbb{E}_{\boldsymbol{Z}_k}[\Omega_k S_k^{(1)}]\right\|_F$$

$$\le \mathbb{E}_{\boldsymbol{Z}_k}\left\|\Omega_k S_k^{(1)}\right\|_F \le \mathbb{E}_{\boldsymbol{Z}_k}\left\|\Omega_k \tilde{S}_k^{(1)}\right\|_F \le \mathbb{E}_{\boldsymbol{Z}_k}\left\|\boldsymbol{G}_k \tilde{S}_k^{(1)}\right\|_F \le G^\alpha (d_k G_4)^{1-\alpha}, \tag{111}$$

Furthermore, we use Assumption (A4) and Lemma B.2 to derive a lower bound for $a_{ij}^{(k)}$ where

$$a_{ij}^{(k)} \geq \frac{mn\epsilon_1^2}{4\mathcal{G}}, \quad \left\| \frac{\bar{G}_k}{\sqrt{A_k}} \right\|_F \leq \frac{\|\bar{G}_k\|_F}{\min_{i,j} \sqrt{a_{ij}^{(k)}}} \leq \frac{2G\sqrt{\mathcal{G}}}{\sqrt{mn}\epsilon_1}. \tag{112}$$

Combining with (99), (110), (111) and (112), we thus derive that

$$\mathbf{D.3} \leq \frac{2G^{1+\alpha}G_4^{1-\alpha}\sqrt{\mathcal{G}}(\epsilon_2 + \Theta_{\max})\rho_0}{\sqrt{mn}\epsilon_1} \sum_{k=1}^{t} \frac{1}{d_k^{\alpha-1}\sqrt{k}}. \tag{113}$$

**Putting together**  Both **E** and **D.1** are bounded with two estimations, one of which owns a better dependency to $1/\epsilon_1$ and the other avoids the curse of the dimension. We thereby derive two results. Plugging (106), (109) and (113) into (101) and then combining with (102) and (100), we then derive that with probability at least $1 - \delta$, for all $t \in [T]$,

$$\frac{1}{2}\sum_{k=1}^{t} \hat{\rho}_k \left\| \frac{\bar{G}_k}{\sqrt[4]{A_k}} \right\|_F^2 \leq C_1 \log\left(\frac{T}{\delta}\right) + C_2 \sum_{k=1}^{t}(1 - \beta_{2,k}) + C_3 + D_1(\alpha)\sum_{k=1}^{t} \frac{1}{d_k^{\alpha-1}\sqrt{k}}, \tag{114}$$

where $C_1, C_2, C_3$ are as in Theorem B.1 and $D_1(\alpha)$ is as in (94). Plugging (107), (109) and (113) into (101), then combining with (104) and (100), we then derive that with probability at least $1 - \delta$, for all $t \in [T]$,

$$\frac{1}{2}\sum_{k=1}^{t} \hat{\rho}_k \left\| \frac{\bar{G}_k}{\sqrt[4]{A_k}} \right\|_F^2 \leq C_1 \log\left(\frac{T}{\delta}\right) + C_2' \sum_{k=1}^{t} \frac{1}{k^{c/2+1/2}} + C_3' \sum_{k=1}^{t} \frac{1}{k} + D_1(\alpha)\sum_{k=1}^{t} \frac{1}{d_k^{\alpha-1}\sqrt{k}}. \tag{115}$$

where $C_2', C_3'$ are as in Theorem B.1. Moreover, using (99), we reveal that the lower bound for $\hat{\rho}_k$ is the same the one for $\eta_k$ in (58). Thereby, following the same deduction in (80) and (86), we derive that

$$\frac{1}{2}\sum_{k=1}^{T} \hat{\rho}_k \left\| \frac{\bar{G}_k}{\sqrt[4]{A_k}} \right\|_F^2 \geq \sum_{k=1}^{T} \frac{\hat{\rho}_k}{2} \frac{\|\bar{G}_k\|_F^2}{\max_{i,j} \sqrt{a_{ij}^{(k)}}} \geq \frac{1}{D_0}\sum_{k=1}^{T} \frac{1}{\sqrt{k}} \|\bar{G}_k\|_F^2, \tag{116}$$

where $D_0 = \min\{C_0, C_0'\}$ that has been defined in (94). Setting $t = T$ on (114) and (115), and then using (116), we then derive that

$$\min_{t \in [T]} \|\bar{G}_k\|_F^2 \leq \frac{D_0}{\sum_{t=1}^{T} 1/\sqrt{k}}\left(C_1 \log\left(\frac{T}{\delta}\right) + C_2 \sum_{k=1}^{t}(1 - \beta_{2,k}) + C_3 + D_1(\alpha)\sum_{k=1}^{t} \frac{1}{d_k^{\alpha-1}\sqrt{k}}\right),$$

$$\min_{t \in [T]} \|\bar{G}_k\|_F^2 \leq \frac{D_0}{\sum_{t=1}^{T} 1/\sqrt{k}}\left(C_1 \log\left(\frac{T}{\delta}\right) + C_2' \sum_{k=1}^{t} \frac{1}{k^{(c+1)/2}} + C_3' \sum_{k=1}^{t} \frac{1}{k} + D_1(\alpha)\sum_{k=1}^{t} \frac{1}{d_k^{\alpha-1}\sqrt{k}}\right).$$

Then, using the results in (88) and (89), we could derive the desired result in Theorem C.1.

