# OpenReview forum: "Convergence of Adafactor under Non-Convex Smooth Stochastic Optimization"
_NeurIPS.cc/2024/Conference — Submitted to NeurIPS 2024_

### Official Review · Reviewer_1HAW · 2024-07-13

**Soundness:** 3
**Presentation:** 2
**Contribution:** 3
**Rating:** 7
**Confidence:** 3

**Summary:**

The paper examines the convergence properties of Adafactor, an adaptive learning rate optimizer designed for deep learning tasks, particularly in memory-constrained environments. The study focuses on Adafactor’s performance in non-convex optimization scenarios and provides theoretical convergence proofs under smooth conditions. Despite its widespread practical use, especially for training large language models, Adafactor’s theoretical understanding has been limited. This research fills that gap by proving that Adafactor can reach a stationary point with a specific convergence rate, highlighting both its efficiency and the impact of different parameter settings on its performance. The paper also introduces modifications to the default hyper-parameter settings based on theoretical insights, which are validated through empirical tests, showing potential improvements over traditional setups.

**Strengths:**

The analysis of Adafactor’s convergence is crucial for its application in training extremely large models, such as large language models (LLMs). This study significantly contributes to the theoretical foundations of Adafactor, supporting its practical use with a solid mathematical framework.

**Weaknesses:**

The manuscript could benefit from a deeper discussion on the proving techniques and the tightness of the provided convergence bounds.

**Questions:**

1. Could you clarify the primary differences or challenges encountered in proving the convergence of Adafactor compared to other adaptive gradient-based methods?

2. The convergence bound suggests that Adafactor is not affected by the curse of dimensionality related to the factor $mn$. Have you observed this phenomenon in practical experiments or applications?

3. Is there potential to improve the current convergence rate of $\log T/\sqrt{T}$ stated in Theorem 5.1? If so, what changes or assumptions might lead to such an improvement?

**Limitations:**

Yes.

---

> ### Author Rebuttal · Authors · 2024-08-06
>
> We thanks a lot for the reviewer's feedback and valuable suggestions. Below are our responses to the major concern.
>
> **Response to Weakness: we will add some deeper discussion per your suggestions (refer to the global rebuttal for details)**
> - We kindly refer the reviewer to see the **global rebuttal** for a brief introduction of our proof challenges and techniques, which will be properly added in the appendix. We will also add proof sketches for each main results in the appendix accordingly. Please also see the proof sketch example in Rebuttal to Reviewer 5LgA.
>
> - We agree that there are some loose terms in our convergence bounds, such as $1/\epsilon_1$ as we discuss in Section 6.1 and the high order of gradient bound $G$. We will leave this interesting topic for the future work.
>
> **Response to Question**
> 1. We refer the reviewer to see the **global rebuttal** where we state several proof challenges and techniques that are quite different to other adaptive methods.
> 2. As far as we can understand, this question refers to whether the model dimension will affect the performance of Adafactor. Considering experiments of Resnet-20 and Resnet-110 on CIFAR-100 in Figure 3, we think that the order of convergence rate are the same since the former uses around 20k steps and the latter around 10k steps. Both models achieve comparable performance (test accuracy) in Figure 1. This experiment somehow supports the phenomenon that Adafactor is not affected by the curse of dimensionality. We are lack of the evidence in NLP field, which shall be further developed in the future.
> 3. If the step-size $\rho_k = \rho_0$, then the theoretical convergence rate could be improved to $\log T /T$, which could be verified through [Line 490] and will be included in the revision. Thanks a lot for your insightful comments. However, we are not sure whether the convergence rate of full-batch Adafactor could be improved to $\log T /T$ under $\rho_k=\rho_0/\sqrt{k}$.

---

> > ### Comment · Reviewer_1HAW · 2024-08-13
> >
> > Thank you for answering my questions. I keep my original score.

---

> > > ### Author Response · Authors · 2024-08-13
> > >
> > > Dear Reviewer 1HAW,
> > >
> > > Thanks a lot again for your positive feedback and support. Based on your comment, we promise to make the following revisions accordingly in the revised version:
> > >
> > > - We will add a paragraph in the appendix to briefly state the proof novelty (as summarized in the global rebuttal).
> > > - We will add a proof sketch for Theorem 6.1 in the main body (as stated in the rebuttal to Reviewer 5LgA) and proof sketches for other main results in the appendix.
> > > - Section 6.1 will be polished and added with some discussion of the tightness of the convergence as follows: "There also remain some loose terms in our convergence bounds. For example, the term $\mathcal{O}(1/\epsilon_1)$ and the high order of $G$. We believe that using the step-size $\rho_0 \sim \mathcal{O}(1/G)$, the order of $G$ may be potentially reduced. The improvement for the order of $1/\epsilon_1$ would be an interesting topic for the future work".
> > > - The experiment in the attached PDF will be included in the main body, serving as a supplementary support for our main results.
> > >
> > > Best regards,
> > >
> > > Authors.

---

### Official Review · Reviewer_ExZW · 2024-07-29

**Soundness:** 3
**Presentation:** 3
**Contribution:** 1
**Rating:** 3
**Confidence:** 4

**Summary:**

This paper studies the convergence of Adafactor for non-convex smooth objectives. The paper looks at both full batch and stochastic cases and analyze the convergence rate. Experiments are provide to validate some of the findings about the hyperparameters.

The main contributions of this paper are: (1) convergence rates for both full-batch and stochastic cases for Adafactor (2) Provide empirical evidence that the hyperparameters leading to optimal convergence rates yields better empirical performance. My main concern regarding this paper is about novelty and lack of comprehensive experimental evidence. First regarding novelty, convergence of Adaptive methods has been studied in several earlier papers for full-batch settings (e.g. De et.al., 2018). Similar, the issue of second moment decay parameter increasing at the rate of 1 - 1/k is well known (e.g. Reddi et al., 2018) and under this particular schedule, the algorithms roughly boils down to Adagrad like schedule (instead of exponential moving average). Both these contributions are quite well-known for adaptive methods. Thus, it is not entirely clear to me if these contributions for Adafactor are novel enough to warrant acceptance.

Furthermore, Shazeer et.al., in Section 7.2 of their paper already discuss about the aspect of second moments decay. The experiments in the current paper are neither comprehensive or convincing that this leads to he optimal convergence in practice. It is important to do a very thorough investigation if the authors have to demonstrate this phenomenon (e.g. try it in different NLP settings where adaptive methods are very effective). Overall, in my opinion, the main weaknesses of this paper are novelty and poor empirical study.

**Strengths:**

See summary

**Weaknesses:**

See summary

**Questions:**

See summary

**Limitations:**

See summary

---

> ### Author Rebuttal · Authors · 2024-08-06
>
> Thanks a lot for you time on our paper and feedback.
>
> **To summarize, Reviewer ExZW has the following two major concerns (Please correct me if I was wrong).**
>
> - **Concern 1**. Lack of novelty for  convergence analysis of Adafactor comparing with other adaptive methods such as Adam  and AMSGrad.
> - **Concern 2**. Poor empirical study.
> -------
> **Response to Concern 1: We politely clarify as follows, and we claim that our analysis for Adafactor is novel.**
>
> - Adafactor is a memory-constrained optimizer while Adam and AMSGrad are not.
> - Most of the analysis for adaptive methods in nonconvex smooth setting are only for memory-unconstrained optimizers such as Adam, our analysis for Adafactor may be the first for
> memory-constrained optimizers involving matrix factorization, to our knowledge.
> - The analysis for Adafactor has major differences with those for Adam and AMSGrad.
> - Adafactor is one of the popular algorithms proposed in 2018, but theoretical analysis is undeveloped to our knowledge, which somewhat is due to its analysis challenging.
> - We summarize the major differences of Adafactor to Adam in three points (the detailed comparison could be found in the attached PDF):
> 	- The unique update-clipping for Adafactor, defined by $1/\max(\text{RMS}(\bar{U}_k),1)$ in the step-size;
> 	- The rank-1 matrix factorization, which leads to the unique adaptive step-size $\bar{W}_k = \frac{\bar{R}_k\bar{C}_k}{\bar{S}_k}$ instead of the expoential moving average matrix $\bar{V}_k$ [Eq. (12), Line 445];
> 	- The time-varying $\beta_{2,k}$ instead of a constant $\beta_2$.
>
> 	These differences lead to several essential proof challenges and techniques in our paper, which are summarized in the global rebuttal. Regarding the specific concerns raised in the comment:
>
> (a). Concern on proof similarity to existing methods (full-batch case) for Adam in (De et al., 2018). We clarify the major proof difference in the global rebuttal. We believe that there exist some essential proof differences to those for Adam in (De et al., 2018).
>
> (b). Concern on the decay parameter been studied for AMSGrad (Reddi et al., 2018). Considering the fundamental differences between Adafactor and AMSGrad, this finding is non-trivial.
>
> ---
> **Response to Concern 2: We clarify as follows, and we will also add some numerical results per your suggestion.**
>
> - Numerical results in different NLP settings have been given in [26] for Adafactor  with its default numerical parameter settings to show its efficiency.  The theoretical analysis for Adafactor is undeveloped to our knowledge.
> - Our paper is primarily a theoretical work, with its main achievement being the first proof that Adafactor can converge (covering the  default numerical parameter settings although with a suboptimal rate, and the other regimes with optimal rate). Our experimental results are supplementary to our theoretical findings and thus were initially kept simple.
>
> - Per your suggestion, in response, we also have included experiments with BERT-Base [Devlin et al., 2018] model on the GLUE/MNLI dataset, and the results have been added to the attached PDF of the global rebuttal.
>
>
> J. Devlin, M.-W. Chang, K. Lee, and K. Toutanova. Bert: Pre-training of deep bidirectional transformers for language understanding, 2018.
>
> ---
>
> Thank you for your comments and the reference [Reddi et al., 2018], which will be incorporated carefully in the revision.

---

> ### Author Response · Authors · 2024-08-14
> **on the similar convergence results between Adafactor and Adam (following-up the above comment)**
>
> Dear Reviewer ExZW
>
> We see that your major concerns are on the theoretical similarity of our results for Adafactor with those for Adam.
> (please kindly correct me if I am wrong and we were sorry for the misunderstanding).
>
> Intuitively, one can not prove better theoretical results of Adafactor than Adam, as Adafactor is a memory-saved variant of Adam.
> Empirical results in [26] show that Adafactor and Adam perform roughly similarly.
>
> According to our results, one can assert that Adafactor and Adam have similar theoretical results, while Adafactor needs less memory than  Adam, which somewhat complements the extensive empirical results in [26].
>
> We hope this could potentially address your concerns and may slightly change your mind on the evaluation.
>
> If concerns remain, please feel free to bring them to our attention. We would be more than happy to discuss them with you.
>
> We look forward to your comments and discussions. Thanks.
>
> authors

---

### Official Review · Reviewer_5LgA · 2024-07-30

**Soundness:** 3
**Presentation:** 2
**Contribution:** 3
**Rating:** 6
**Confidence:** 3

**Summary:**

This paper studies the convergence of a memory-efficient, adaptive algorithm, Adafactor, under non-convex smooth settings. First, the authors show that in the full-batch setting (with appropriate hyperparameters), Adafactor converges to a stationary point at an $\tilde{O}(1/\sqrt{T})$ rate. For the stochastic setting, they study two regimes: with and without clipping $\eta_k$, and show that under appropriate selection of hyperparameters, Adafactor attains an $\tilde{O}(1/\sqrt{T})$ rate of convergence to the stationary point, matching SGD up to logarithmic factors. The observations are complemented with empirical findings.

**Strengths:**

Considering large-scale language model training, the work focuses on an important memory-constrained practical optimizer for which we have limited theoretical understanding. The work is clearly motivated, the introduction to Adafactor and its connection to Adam is concisely discussed, and the paper is easy to follow in general. Even though it’s not exactly vanilla Adafactor, it’s exciting to see the authors have established bounds matching SGD in the stochastic case. I also appreciate the authors not shying away from discussing the impact of various hyperparameters (and potential negative points).

**Weaknesses:**

Even though there are space constraints, I would like to see some proof sketch (at least for the full-batch case) in the main body to provide a general sense for the reader. For example, it could be as simple as starting from smoothness in Taylor series (Eq. 14), telescoping over $k$, lower bounding (a), upper bounding (b) in Eq 20. The more critical step appears to be the lower bound, and a general flavour of how Lemmas $A.2, A.3$ are used to achieve that would be nice to see.

This is minor, but it would be helpful for the reader if figures are referenced whenever a discussion about some experiment is invoked. Two instances I noticed are line 222, the effect of $\epsilon_1$, and line 265, the effect of time-increasing $d_k$. Please look for other instances, if any.

Figure 1, experiment on the effect of different decay rate parameter $c$: Showing the test performance is nice, but as the discussion is about train-loss convergence, their time-evolution should be included. It’s fine if the convergence doesn’t speed up with increasing ($c$) and doesn’t match theory, but its important to have them as the entire work deals only with train-loss convergence.

**Questions:**

On Assumption A4: Did you try considering the less restrictive expectation bound and establishing expectation convergence results instead of high-probability results for Thm. $6.1$, $7.1$? I am curious to hear your thoughts on this.

On Thm $5.1$: Why you think this bound is loose? What part of the analysis is preventing the establishment of at least the $O(1/T)$ rate?

Line 178: Under learning rate warmup, update clipping shows little effect on performance, and this is used as one of the reasons to drop update clipping. However, in Thm $6.1$, I see no warm-up on $\rho_k$, it just decreases with $k$. Please correct me if I am misunderstanding something. If not, I don’t mind analysing the a slightly modified algorithm for theory, but the reasons should justify the choices made.

**Limitations:**

Yes, the authors have addressed the limitations.

---

> ### Author Rebuttal · Authors · 2024-08-06
>
> We thanks a lot for the reviewer's effort invested on our paper.
>
> **Response to Weakness: We will revise accordingly per your suggestions on the presentation issue.**
> - We present a proof sketch for Theorem 6.1 (the stochastic case) as follows. The proof sketches for other main results will also be included in the revised version. We begin by the descent lemma of the smoothness and introduce $A_k$ defined in [Eq. (50), L531]. Then, summing up both sides over $k \in [t]$,
> $$
> f(X\_{t+1}) \le f(X\_{1})\underbrace{ - \sum\_{k=1}^t\eta\_k \left\langle \bar{G}\_{k}, \frac{G\_k}{\sqrt{A\_k}} \right\rangle}\_{\bf A} + \underbrace{\sum\_{k=1}^t\eta\_k \left\langle \bar{G}\_{k}, G\_k\odot\left(\frac{1}{\sqrt{A\_k}}-\frac{1}{\sqrt{W\_k}} \right)\right\rangle}\_{\bf B} + \underbrace{\sum\_{k=1}^t\frac{L\eta\_k^2}{2}\left\\|\frac{G\_k}{\sqrt{W\_k}}\right\\|\_{F}^2}\_{\bf C}.
> $$
> We introduce $\xi\_k$ into **A** and then use the concentration inequality of the martingale difference sequence, leading to with probability at least $1-\delta$,
> $$
> {\bf A} \preccurlyeq -\frac{3}{4}\sum\_{k=1}^t \eta_k \left\\|\frac{\bar{G}\_k}{\sqrt[4]{A\_k}} \right\\|\_F^2+ \mathcal{O}\left(\frac{G^2\log (T/\delta)}{\epsilon\_1}\right).
> $$
> Relying on the delicate construction of $A_k$, we are able to control the relative distance of $W_k$ and $A_k$ (detailed in Lemma B.7)
> $$
> \frac{\left|w\_{ij}^{(k)} - a\_{ij}^{(k)}\right|}{\sqrt{ a\_{ij}^{(k)}}} \preccurlyeq  \mathcal{O}\left(G\sqrt{1-\beta\_{2,k}}\right)\quad \forall k \ge 1,i\in[n],j\in[m].
> $$
> Thereby, we could control **B** using the above result and some basic inequalities,
> $$
> {\bf B} \preccurlyeq  \frac{1}{4}\sum\_{k=1}^t \eta\_k \left\\|\frac{\bar{G}\_k}{\sqrt[4]{A\_k}} \right\\|\_F^2  + \mathcal{O}\left(G\sum\_{k=1}^t (1-\beta\_{2,k})\left\\|\frac{G\_k}{\sqrt{W\_k}} \right\\|\_F^2\right).
> $$
> In order to bound the sum of the second-order term emerged in **B** and **C**, we first control the ratio of the second-order term for Adafactor and Adam. Then, we generalize an inequality related to the sum of second-order term for Adam with constant decay $\beta\_2$ [Lemma 5.2, 8] to a time-varying setup. These results are summarized as follows (see also Lemma B.4 and B.5),
> $$
> \left\\|\frac{G\_k}{\sqrt{W\_k}} \right\\|\_F^2 \preccurlyeq \mathcal{O}\left(\frac{G^2}{\epsilon\_1}\left\\|\frac{G\_k}{\sqrt{V\_k}}\right\\|\_F^2\right), \quad \sum\_{k=1}^t (1-\beta\_{2,k})\left\\|\frac{G\_k}{\sqrt{V\_k}} \right\\|\_F^2 \preccurlyeq \mathcal{O}\left( \log\left({G^2 \over \epsilon\_1} \right) + \sum\_{k=1}^t (1-\beta\_{2,k})\right).
> $$
> Noting that $\eta_k^2 \preccurlyeq \mathcal{O}(1/k) \preccurlyeq \mathcal{O}(1-\beta_{2,k})$ when $\beta_{2,k}=1-1/k^c, c\in [1/2,1]$. Then, we derive from the above two results that
> $$
> {\bf B+C} \preccurlyeq \frac{1}{4}\sum\_{k=1}^t \eta\_k \left\\|\frac{\bar{G}\_k}{\sqrt[4]{A\_k}} \right\\|\_F^2  + \mathcal{O}\left(\frac{G^3}{\epsilon\_1}\left(\log\left({G^2 \over \epsilon\_1} \right) + \sum\_{k=1}^t (1-\beta\_{2,k})\right)\right).
> $$
> Plugging the bounds for **A,B,C** into the first inequality, we then derive that with probability at least $1-\delta$,
> $$
> \frac{1}{2}\sum\_{k=1}^t \eta\_k \left\\|\frac{\bar{G}\_k}{\sqrt[4]{A\_k}}\right\\|_F^2 \preccurlyeq \mathcal{O}\left(\frac{G^3}{\epsilon\_1}\left( \log\left(\frac{GT}{\delta\epsilon\_1}\right) + \sum\_{k=1}^t (1-\beta\_{2,k})\right) \right).
> $$
> Further upper bounding $\\|\sqrt[4]{A_k}\\|\_F$ and using some simple calculation, we are able to prove the final desired result.
> - Here are the instances where we will add the corresponding reference for figures (including two instances the reviewer mentioned): [L197, Figure 1], [L212, Figure 1], [L222, Figure 2], [L253/L265, Figure 3 and 4].
> - We add the training loss figure for Experiment 1 into the attached PDF.
>
> **Response for Question**\
> Thank you for your insightful comments and detailed readings.
> - The expected result is interesting, but we do not consider at this moment. We conjecture that the expected bounded gradient assumption, $\mathbb{E}\\|G_k\\|\le G$, may lead to an expected convergence. However, there may exist several challenges and differences compared to the high probability result. For example, the difference may arise in lower bounding the LHS of [Eq. (86), L643], which may lead to a sub-optimal rate. We are happy and open if you would like to discuss more.
> - The reason may come in two points:
>   - the time-varying step-size $\rho_k$. When setting $\rho_k=\rho_0$ in [L490], we are able to derive the optimal rate.
>   - It's noticed that [28] present the optimal rate for full-batch RMSProp and Adam with $1/\sqrt{k}$ step-size, relying on the exponential moving average property. However, it seems unknown on how to lower bound **(a)** [Eq. (20), L476] more tightly, given the unique adaptive step-size and update-clipping in Adafactor.
> - We mainly consider the stage when the warm-up is finished. In the other word, the initial point $X_1$ is the output of the warm-up stage. This allows us to simplify our analysis and focus on the stage that leads to the final output. We also believe that investigating warm-up stage could be a quite interesting topic for the future work.

---

> ### Comment · Reviewer_5LgA · 2024-08-13
>
> Hi,
>
> Thanks for answering my questions, and the detailed responses in the rebuttal. Adding the proof sketch would be a nice addition to the paper (both full and stochastic batch would be nice, but if not at least the full). I'd also like that you like the Fig. 1 from the PDF (if you cannot fit in paper, do the App. and reference it, please). Could you also discuss the optimality of Thm 5.1 and that setting $\rho_k = \rho_0$ gives you the optimal bound in the paper? Considering the rebuttal, I will maintain my positive rating of this work.

---

> ### Author Response · Authors · 2024-08-13
> **Thanks a lot for your positive reply!**
>
> Dear Reviewer 5LgA,
>
> Thanks a lot for your detailed reading and technically insightful comments. In response, we will include the following materials in the main body or in the appendix of the revised version.
>
> - We will add the proof sketches for all of our results. We will include the sketch for the stochastic case in the main body if space allows. The other sketches will be included in the appendix.
>
> - We will include the experiment result in the attached PDF into the main body.
>
> - To discuss the the optimality of Thm 5.1 and the case of $\rho_k = \rho_0$, we will add the result under $\rho_k = \rho_0$ into Thm 5.1 and  revise the paragraph after Thm 5.1 (L169-L173) as follows:
>
> "The result indicates that full-batch Adafactor could find a stationary point at a rate of $\mathcal{O}(1/\sqrt{T})$ under the non-convex smooth case. This rate is similar to gradient descent but with a sub-optimal rate $\mathcal{O}(1/\sqrt{T})$ compared to $\mathcal{O}(1/T)$ for SGD [4] and Adam [28] under the setup $\rho_k \sim \mathcal{O}(1/\sqrt{k})$. It was noticed that when $\rho_k$ is set as the constant $\rho_0$, the rate is improved to $\mathcal{O}(1/T)$.
>
>   It remains uncertain whether the convergence rate of full-batch Adafactor can be improved to the optimal rate under the default setup. We believe the sub-optimal rate may stem from Adafactor's unique structure, which involves update-clipping and deviates from the exponential moving average used by Adam. These factors create challenges in lower bounding the first-order term in the descent lemma, resulting in a sub-optimal rate. Improving this to an optimal rate would be an interesting direction for future work."
>
> ---
>
>   We sincerely thank Reviewer 5LgA's positive valuable comment and support. lf you feel it's appropriate, any further consideration, including a possible score adjustment, would mean a lot to us.
>
> Best regards,
>
> Authors.

---

### Author Rebuttal · Authors · 2024-08-06

In this general rebuttal, we have
- **clarification on major contribution and proof novelty**
- **formulation comparisons between Adafactor and Adam, and  extra experiments in the attached PDF**.

-----

**The contribution of our paper could be enough to warrant acceptance:**

- Introduced by [26] that currently has 868 citations on Google Scholar, Adafactor is designed as a memory-efficient alternative to Adam. It reduces memory usage by using the rank-1 factored form and involving other 'clipping and normalizing' steps. Despite its practical success, it lacks any theoretical analysis to the best of our knowledge.
- Our theoretical results in this manuscript may be the first to demonstrate convergence for Adafactor to the best of our knowledge.
- Our proof overcomes several mathematical challenges in comparison with the existing analysis for memory-unconstrained adaptive methods such as Adam and AMSGrad, which we will elaborate in details as follows. Additionally, we have included a comparison of the algorithm forms in the attached PDF.

---

**Proof challenge 1. Lower bound first-order term in the descent lemma (full-batch case)**

In full-batch analysis, since RMSProp/Adam applies the exponential moving average on $\bar{V}_k$, existing results e.g., [7] could rely on gradient bound $G$ and obtain that
$$
\bar{v}\_{ij}^{(k)} \le (1-\beta\_2)\sum\_{l=1}^k\beta\_2^{k-l}G^2 = G^2(1-\beta\_2^k), \forall i \in [n], j \in [m],
$$
thus further bounding **(a)**. However, $\bar{W}\_k$ in Adafactor does not enjoy the exponential moving average. In addition, we should consider the effect of update-clipping.

**Solution.** We first present two results for $\bar{V}\_k$ (defined in Adam):
- its row/column/coordinate sum matrix $\bar{R}\_k,\bar{C}\_k,\bar{S}\_k$ are updated by the exponential moving average rule (Lemma A.2);
- the coordinate lower/upper bounds for $\bar{R}\_k,\bar{C}\_k,\bar{S}\_k$ are bounded by $\epsilon\_1$ and $G$ (Lemma A.3).

Based on two results, we could upper bound $\bar{W}_k$ in [Eq. (28), L487] and [Eq. (34), L496] using some basic inequalities. In addition, we handle the update-clipping by considering two cases and further lower bound **(a)** in [Eq. (20), L476].

---
**Proof challenge 2. A new adaptive step-size (stochastic case)**

In the stochastic case, there are two central and unique challenges in the analysis of Adafactor, both of which are brought by the adaptive step-size involving an unique matrix factorization:
- to handle the entanglement of stochastic gradient $G_k$ and the adaptive step-size $W_k$;
- to control the summation of the second-order term $\sum_{k=1}^t\\|\frac{G_k}{W_k}\\|_F^2$.

**Proof challenge 2.1**. In the first challenge, a common way is to construct a proxy step-size to break the entanglement. However, note that
$$
\text{Adam/RMSProp:}\quad V\_k = \beta\_2 V\_{k-1}+(1-\beta\_2) G\_k^2, \quad \text{AdaGrad:}\quad V\_k = V\_{k-1} + G\_k^2.
$$
$V_k$ and $V_{k-1}$ share a linear relation in Adam/RMSProp/Adagrad. Then, existing types of proxy step-size e.g., [32, 8] use some decorrelated terms to replace $G_k^2$. However, since there is **no linear relation** between $W_k$ and $W_{k-1}$ in Adafactor, it's unknown that whether this solution still works.

**Solution.** We reveal that (see the detail in [Eq. (48), L529, Eq. (49), L530]))
$$
\text{Adafactor:}\quad W_k = \frac{(\beta_{2,k} R_k+(1-\beta_{2,k})R_{G^2_k})(\beta_{2,k} C_k+(1-\beta_{2,k})C_{G^2_k})}{(\beta_{2,k} S_k+(1-\beta_{2,k})S_{G^2_k})}.
$$
We replace $R_{G_k^2},C_{G_k^2},S_{G_k}^2$ with the decorrelated terms related to $G$. This leads to a new proxy step-size $A_k$ in [Eq. (50), L531]. We further find that the relative distance of $W_k,A_k$ (the target term in Lemma B.7) is bounded by $\mathcal{O}(G\sqrt{1-\beta_{2,k}})$, which is the key step to control the error term **B** [Eq. (70), L621]. The deduction in Lemma B.7 is non-trivial and requires some more new estimations compared to those for AdaGrad [Eq. (7,8,9), 32] and Adam [Lemma 5.1, 8].

**Proof challenge 2.2.** To overcome the second challenge, one of a common way is to apply [Lemma 3.2, 32] for AdaGrad and [Lemma 5.2, 8] for Adam. However, these results could not be directly applied to Adafactor due to the different adaptive step-size and the time-varying $\beta_{2,k}$.

**Solution.** We first control the ratio of the second-order term for Adafactor and Adam in Lemma B.4. Then, we extend [Lemma 5.2, 8] to the time-varying $\beta_{2,k}$ (Lemma B.5). Combining with two lemmas, we are able to show the sum of the second-order term is bounded by a logarithmic term.

---
**Proof challenge 3. Update-clipping in the stochastic case**

We first emphasize that the additional update-clipping leads to a rather complicated adaptive step-size $\tilde{W}_k$ in Adafactor,
$$
\tilde{W}\_k = \frac{W\_k}{\max\\{1,\\|{U}\_k\\|\_F/(d\_k\sqrt{mn})\\}}, \quad \text{where} \quad W\_k = \frac{R\_kC\_k}{S\_k},U\_k=\frac{G\_k}{W\_k}.
$$
To our knowledge, this complicated structure causes all existing constructions of proxy step-size, including our setup mentioned in the second point, to fail.

**Solution.** Inspired by a standard way in the analysis of SGD with clipping, we provide a decomposition in [Eq. (101), L678]. The central idea is to incorporate the update-clipping into stochastic gradient $G_k$ and define a new term $\tilde{G}_k$ [Eq. (98), L673]. Relying on gradient bound $G$ and the definition of the update-clipping, we are able to control the key error term **D.3** in [Eq. (113), L707]. Consequently, we should further require the clipping threshold $d_k = k^{\frac{c}{2(\alpha-1)}}$.

---
We will consider adding the above statements in the appendix.

---

### Decision · Program_Chairs · 2024-09-25

**Decision:**

Reject

**Comment:**

This paper investigates the convergence of Adafactor in non-convex settings. Adafactor is widely used for memory constrained optimization settings, so its convergence analysis is useful. However, I agree with Reviewer ExZW that the paper does not provide any novel insights either theoretically or empirically to justify its acceptance in the conference. The only empirical insight from the analysis is the role of second moment decay parameter, which, as the reviewer rightly pointed out, has already been extensively studied in the adaptive methods literature. The convergence result itself is not very insightful either providing no evidence why Adafactor can be an effective optimization algorithm. I recommend rejection in the current form and encourage the authors to address these concerns.